# Mechanism of regulation of the *Helicobacter pylori* Cagβ ATPase by CagZ

Xiuling Wu[1,7,8], Yanhe Zhao[2,8], Hong Zhang[1], Wendi Yang[1], Jinbo Yang[1], Lifang Sun[1], Meiqin Jiang[1], Qin Wang[3], Qianchao Wang[1], Xianren Ye[4], Xuewu Zhang [5,6] ✉ & Yunkun Wu [1] ✉

The transport of the CagA effector into gastric epithelial cells by the Cag Type IV secretion system (Cag T4SS) of *Helicobacter pylori* (*H. pylori*) is critical for pathogenesis. CagA is recruited to Cag T4SS by the Cagβ ATPase. CagZ, a unique protein in *H. pylori*, regulates Cagβ-mediated CagA transport, but the underlying mechanisms remain unclear. Here we report the crystal structure of the cytosolic region of Cagβ, showing a typical ring-like hexameric assembly. The central channel of the ring is narrow, suggesting that CagA must unfold for transport through the channel. Our structure of CagZ in complex with the all-alpha domain (AAD) of Cagβ shows that CagZ adopts an overall U-shape and tightly embraces Cagβ. This binding mode of CagZ is incompatible with the formation of the Cagβ hexamer essential for the ATPase activity. CagZ therefore inhibits Cagβ by trapping it in the monomeric state. Based on these findings, we propose a refined model for the transport of CagA by Cagβ.

*H. pylori* is a pathogenic microorganism that colonizes the gastric mucosa of 50% of the world's population, causing various diseases, including peptic ulcer disease, gastric mucosa-associated lymphoid tissue lymphoma and gastric cancers[1–3]. The presence of Cag T4SS, the type IV secretion system encoded by the cytotoxicity-associated gene (Cag) pathogenicity island, is directly associated with the pathogenicity of *H. pylori* strains[4,5]. The pathogenic strains of *H. pylori* use Cag T4SS to deliver the effector protein CagA to the host cells, resulting in altered host cell gene expression, cytoskeleton reorganization, cell morphology changes and inflammatory responses[4–12]. The translocation of CagA itself can increase significantly the risk of gastric cancer[13,14]. CagA is tyrosine-phosphorylated on the EPIYA motif by the Src and Abl kinases in the host cell, interfering with phosphorylation events of the normal signaling cascades[6,15–19]. The effect of phosphorylated CagA is also mediated by its interactions with many host proteins, which have been extensively studied[20]. However, the mechanisms of CagA recruitment and translocation by Cag T4SS remain poorly understood.

T4SSs are transmembrane protein machineries used by various bacteria for horizontal DNA transfer to other bacteria or effector protein translocation to host cells[21–24]. The multiprotein assembly of T4SSs is anchored to both the bacterial inner and outer membranes, and in some cases includes an extracellular pilus as well. The VirB/D system of *Agrobacterium tumefaciens* (*A. tumefaciens*), considered the prototypical T4SS, consists of 12 main constituent proteins, VirB1 to VirB11 and VirD4[25]. VirB4, VirB11 and VirD4, associated with or located in the inner-membrane, are ATPases that provide energy for the secretion of substrates[26,27]. VirD4 plays the role of the coupling protein in T4SS, responsible for the recognition and transport of the substrate to the T4SS channel by using energy released from ATP hydrolysis

[1]Provincial University Key Laboratory of Cellular Stress Response and Metabolic Regulation, Fujian Key Laboratory of Developmental and Neural Biology, Key Laboratory of Optoelectronic Science and Technology for Medicine of Ministry of Education, College of Life Sciences, Fujian Normal University, Fuzhou 350117 Fujian, China. [2]Department of Cell Biology, University of Texas Southwestern Medical Center, Dallas, TX 75390, USA. [3]Department of Biochemistry and Molecular Biology, Binzhou Medical University, Yantai 264003 Shandong, China. [4]Fujian Cancer Hospital & Fujian Medical University Cancer Hospital, Fuzhou 350014 Fujian, China. [5]Department of Pharmacology, University of Texas Southwestern Medical Center, Dallas, TX 75390, USA. [6]Department of Biophysics, University of Texas Southwestern Medical Center, Dallas, TX 75390, USA. [7]Present address: Department of Pharmacology, University of Texas Southwestern Medical Center, Dallas, TX 75390, USA. [8]These authors contributed equally: Xiuling Wu, Yanhe Zhao. ✉e-mail: xuewu.zhang@utsouthwestern.edu; wuyk@fjnu.edu.cn

catalyzed by its ATPase activity[1,28–31]. Crystal structures of TrwB, a homolog of VirD4 in *Escherichia coli* (*E. coli*), show that it is a RecA-like ATPase and forms a ring-like hexamer, characteristic of this type of ATPases[32]. However, a previous structural analysis using negative-stain electron microscopy has proposed a model in which two copies of the VirD4 dimer locate at two sides of T4SS apparatus[33], suggesting that the oligomerization state of VirD4 and related ATPases may be subjected to regulation. The structure of TrwB also revealed an all-alpha domain (AAD), formed by a ~150-residue segment inserted in the middle of the nucleotide-binding domain of the ATPase[32]. The AAD sits at the entrance of the central channel of the TrwB hexamer, which is presumably the transport path for the DNA substrate[32]. It has been shown that the AAD of the VirD4 homologs are involved in substrate recognition, consistent with its location in the structure[34]. The sequence of the AADs of the VirD4 homologs from different species are highly diverse, enabling them to specifically recruit their respective substrates[34].

Cagβ, also known as HP0524 and Cag5, is the homolog of VirD4 in *H. pylori*. Cagβ is essential for recognition and host transport of CagA by Cag T4SS in *H. pylori*, as deletion of Cagβ has been shown to abolish the secretion of CagA[10,35]. As a result, various proteins regulate CagA secretion by interacting with CagA or Cagβ directly. For example, CagF plays a critical role in CagA translocation by acting as a chaperone of CagA[36]. Another interesting regulator is CagZ, a unique protein in *H. pylori*[37]. CagZ has been shown to directly interact with Cagβ and is required for its stability, as strains carrying CagZ mutants produce virtually no Cagβ[35]. Genetic knockout of CagZ abolishes CagA transport to the host cell, underscoring the essential role of CagZ in effector secretion of Cag T4SS[10]. The structure of CagZ shows an L-shaped helical domain followed by a disordered C-terminal tail, which has been proposed to serve as the C-terminal signal for recognition by Cagβ[38].

To understand how Cagβ carries out its function as the coupling protein for CagA and how Cagβ is regulated by CagZ, we determined the crystal structure of the full-length cytosolic region of Cagβ and the Cagβ-AAD in complex with CagZ. Our structural and biochemical data together show that Cagβ functions as a dynamic hexameric assembly. In addition, we show that the binding of CagZ traps Cagβ in the monomeric state, and thereby suppressing its ATPase activity. We propose that CagZ can stabilize the Cagβ protein in the monomeric inactive state, but releases it for the assembly of the hexamer when CagA is ready to be translocated through Cag T4SS to the host cell.

## Results
### The structure of the Cagβ soluble region
We crystallized the soluble region of Cagβ (residues 166–748; referred to as Cagβ_I) and collected an X-ray diffraction dataset to 2.8 Å resolution (Fig. 1a, Supplementary Table 1). Our attempts to solve the structure by molecular replacement using the structures of TrwB (PDB ID:1E9R) and other similar ATPases as the search model failed, suggesting a large structural difference between Cagβ and TrwB. We next generated a model of Cagβ by using AlphaFold 2 in ColabFold[39,40]. Molecular replacement using this model found all six protomers in the asymmetric unit of the crystal, which together form a pseudo-6-fold symmetric hexameric assembly (Fig. 1d–f). Structures of the individual protomer are very similar to each other, with root mean square deviations (r.m.s.d.) below 1 Å. Consistent with the molecular replacement results, the structure of Cagβ is highly similar to the AlphaFold model (r.m.s.d. 1.2 Å), but deviates from that of TrwB substantially (r.m.s.d. 7.3 Å) (Supplementary Fig. 1a, b). The major differences between the Cagβ and TrwB are the peripheral structure elements and the interdomain orientations, while the nucleotide binding domains (NBDs) of the two proteins are quite similar (Supplementary Fig. 1b).

The protomer of Cagβ shows a 3-layered overall architecture similar to TrwB. At the cytosolic end, the AAD (residues 300–489)

forms the top layer of the Cagβ structure (Fig. 1a, b, e, Supplementary Fig. 1c). The AAD in Cagβ contains nine α-helices and two short β-strands. The orientation of the AAD relative to the NBD in Cagβ varies among the six subunits, suggesting some degree of inter-domain flexibility (Fig. 1b, right panel, 1d, inset panel). The NBD, located at the middle layer, adopts the α/β/α-sandwich fold characteristic of the RecA family ATPases (Fig. 1b). Residues 631–658, an extended segment between two α-helices in the NBD, form two long anti-parallel β-strands that stray away from the NBD and constitute the bottom layer of the 3-layered Cagβ (Fig. 1b, Supplementary Fig. 1c). These extended β-strands from the six subunits together form a 12-stranded β-barrel, contributing to the hexameric assembly (Fig. 1e, Supplementary Fig. 1c). A similar β-strand structure is also present in TrwB, although those strands are shorter and oriented differently (Supplementary Fig. 1d). The C-terminal extension after the NBD in Cagβ (residue 687-end), consisting of a three-helix bundle followed by an extended segment, packs against the NBD and likely contributes to its structural stability (Fig. 1a, b, e).

### The hexameric assembly of Cagβ
The hexameric assembly of Cagβ displays the same overall ring-shape as that of TrwB, but the Cagβ ring is much taller because its longer bottom-layer β-barrel and the larger top-layer AAD (Fig. 1e, Supplementary Fig. 1c, d). As in other hexameric ATPases, the NBD makes a major contribution to the inter-subunit interface for the formation of the hexamer, resulting in the placement of the ATP-binding site at the inter-subunit interface (Fig. 1h). The bottom-layer β-strands adopt a highly twisted conformation to form the 12-stranded β-barrel, thereby stabilizing the hexamer through inter-strand hydrogen bonds (Fig. 1i). Near the junction between the β-barrel and the NBD layer, β-strands from neighboring subunits are separated, creating a cavity between the subunits. This cavity is filled by two hydrophobic resides, L183 and F184, from the N-terminal end of the soluble region of Cagβ. In addition, D181 and D182 in the N-terminal segment make charge-charge interactions with R634 and R657 from the neighboring subunit (Fig. 1i). These interactions by the N-terminal segment likely also contribute to the stability of the hexamer. In contrast, the AADs at the top layer make few inter-subunit interactions in the hexamer (Fig. 1g). The very N-terminal portion of Cagβ (residues 1–165), not included in our crystallization construct, is predicted to form three transmembrane helices and a periplasmic loop (residues 91–146)[41], which anchor the Cagβ to the inner membrane of *H. pylori*. Based on the position of the N-terminal segment near the β-barrel in our structure, transmembrane helices likely restrain the β-barrel near the membrane surface. This model would place the AAD distal to the membrane, consistent with its role in recruiting CagA in the cytosol.

The six subunits in the Cagβ hexamer are not related by a perfect 6-fold symmetry. The deviation from the 6-fold symmetry is evident from the top view showing that the distances between neighboring AADs are different from one another substantially (Fig. 1d and inset panel). The distances between the neighboring NBDs also differ, but to a less degree. Similar deviation from 6-fold symmetry has also been seen in TrwB and other hexameric ATPases. It has been established that these ATPases translocate their substrates in a stepwise manner, with different subunits adopting different states at any given time[32,42,43]. The asymmetry in the Cagβ hexamer structure may reflect its intrinsic dynamic nature and ability to carry out sequential translocation of CagA through the channel at the middle of the hexamer.

The central channel in the Cagβ hexamer runs continuously from the cytosol end to the membrane-proximal end, spanning a distance of ~100 Å (Fig. 1c). The channel entrance is relatively narrow, with a diameter of ~9 Å, owing to the protrusion of the α5-α6 loop in the AAD toward the center of the channel (Fig. 1c). The middle portion of the channel formed by the NBD is wider, with the diameter of ~12 Å along its whole length. The channel ends at its membrane side with an

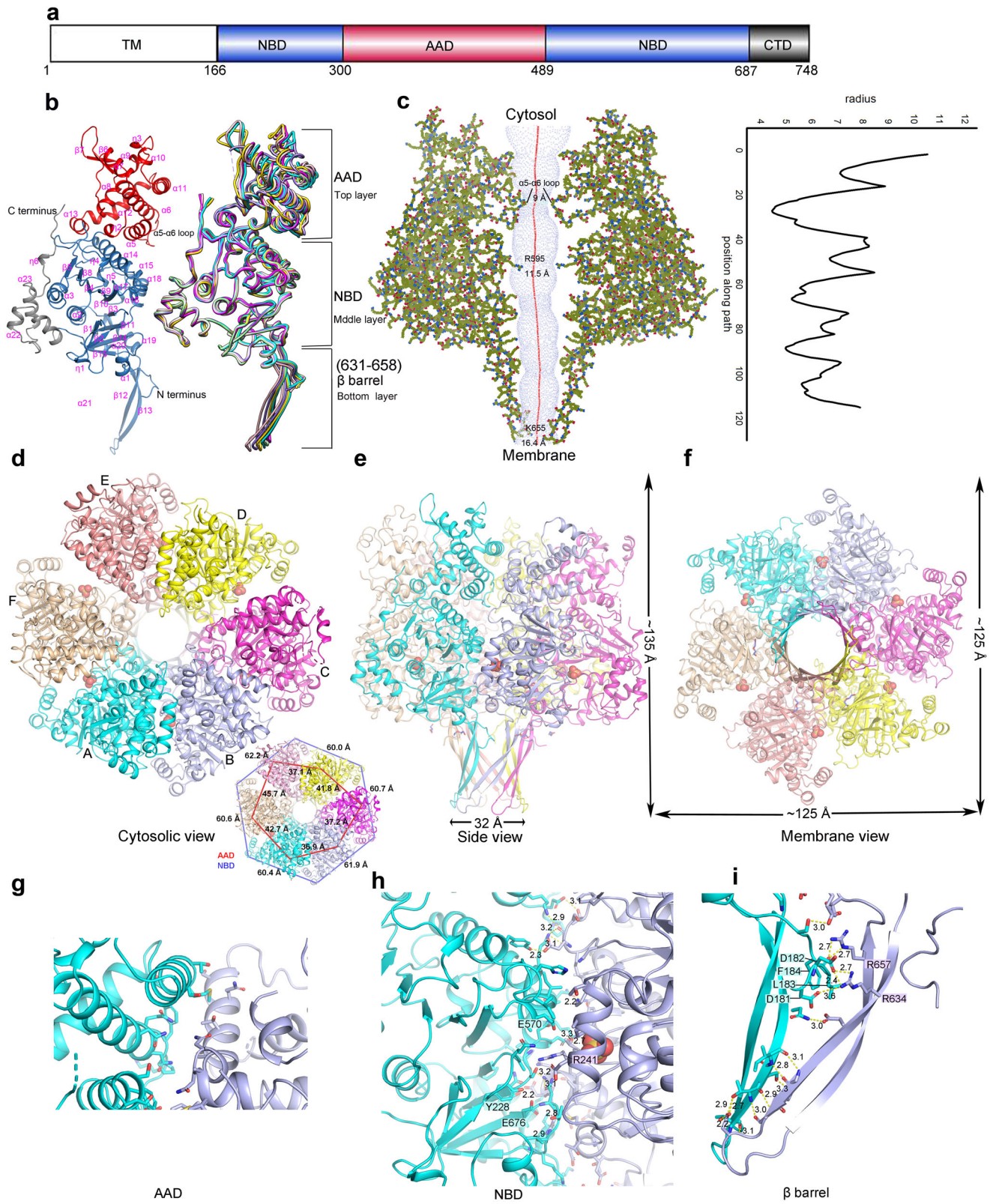

opening of ~16.4 Å. TrwB has been proposed to translocate its DNA substrate through the central channel[32,42]. The surface of the TrwB hexamer shows negative and positive electrostatic potential at its cytosolic and membrane end, respectively, which may help steer its DNA substrate through the central channel toward the membrane (Supplementary Fig. 1e). The surface of the Cagβ does not display such

strong electrostatic polarization, consistent with its role in translocation of the CagA protein rather than DNA (Supplementary Fig. 1f). Nevertheless, there is a small negatively charged patch at the top of the AAD domain (Supplementary Fig. 1f, left panel), which may be involved in interacting with the C-terminal translocation signal in CagA that contains multiple positively changed residues[44].

**Fig. 1 | Structure of the apo-Cagβ hexamer. a** Schematic domain organization of Cagβ. TM the transmembrane region, NBD the nucleotide binding domain, AAD the all-alpha helix domain, CTD the C-terminal domain. Domains included in the crystallization constructs are colored. **b** Ribbon diagram of the apo-Cagβ protomer. The right panel shows a superimposition of six protomers in the hexamer. The protomer in the left panel is colored as in **a**. The color scheme for six protomers in the right panel is as follows: protomer A, cyan; protomer B, light blue; protomer C, magenta; protomer D, yellow; protomer E, pink; protomer F, wheat. **c** Central pore of the hexamer. Left panel, hole profile of Cagβ hexamer. Right panel, pore radius plot of the apo-Cagβ hexamer. **d**–**f** Overviews of the hexamer. The inset shows the

distances between the neighboring subunits in the hexamer. The inter-subunit distances for the AAD (red lines) and the NBD (blue lines) were measured by using residue 374 in the AADs and residue 704 in the NBDs, respectively. These distances together show that the six NBDs are related by an approximate 6-fold symmetry, whereas the six AADs deviate substantially more from this symmetry. **g** Interface between the AADs of protomers A and B. It is evident that these two AADs make minimal inter-subunit contacts. Other AADs in the hexamer are even further apart from one another and therefore do not contribute to the stability of the hexameric assembly. **h, i** The NBD and β-barrel make extensive inter-subunit interactions for the formation of the hexamer.

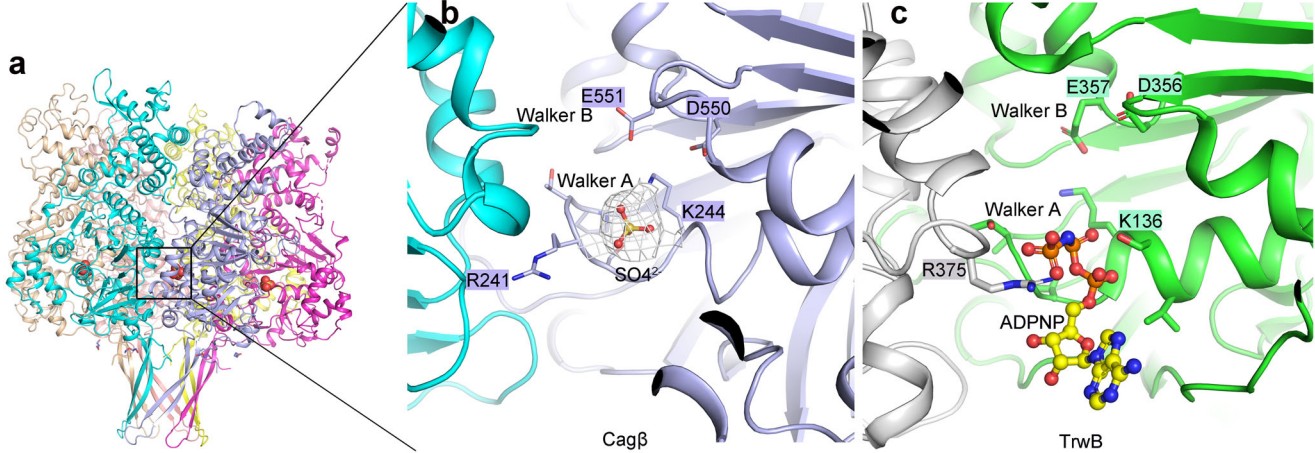

**Fig. 2 | The nucleotide-binding site of Cagβ. a** Side view of the Cagβ hexamer. **b** The nucleotide-binding site of Cagβ. A $SO_4^{2-}$ ion is bound to the activate site, mimicking the ATP substrate. The 2fo-fc density for the $SO_4^{2-}$ group is shown at 1σ

in white. **c** The nucleotide-binding site of TrwB shown in the same orientation for comparison.

## The ATP binding site of Cagβ

A comparison of the NBDs of Cagβ and TrwB reveals the Walker A and Walker B motifs in Cagβ, which are conserved structural elements in these ATPases required for ATP binding and hydrolysis (Fig. 2 and Supplementary Fig. 2a). The Walker A motif is formed by the $^{239}$PTRSGK$^{244}$ loop, in which G243 and K244 are conserved in Walker A in TrwB and other ATPases of the same family. D550 and E551 located in a loop spatially close to the Walker A motif constitute the Walker B motif in Cagβ. Strong density is present near the Walker A and B motifs in the structure, which was assigned to a sulfate group from the crystallization buffer, mimicking the binding of ATP to the nucleotide-binding site (Fig. 2b). The ATP hydrolysis activity of the RecA family ATPases usually relies on the formation of the hexamer, because the catalytically critical arginine finger residue is provided in trans by the neighboring subunit in the hexamer[32,45,46]. As shown in Fig. 2c, the arginine finger in TrwB is Arg375[32,47]. Surprisingly, no arginine residue from the neighboring subunit is present near the ATP-binding site in the Cagβ hexamer (Fig. 2b). Arg241 in the Walker A motif of Cagβ, however, appears to take the same position as the trans Arg375 in TrwB (Supplementary Fig. 2a, b). These analyses suggest that Arg241 in Cagβ may assume the role of arginine finger in cis for catalyzing ATP hydrolysis. The Walker A motif directly interacts with the neighboring Cagβ subunit, suggesting that the formation of the hexamer contributes to the ATPase activity by stabilizing the conformation of the Walker A motif (Fig. 1h). The sidechain of Arg241 makes contacts with several residues from the neighboring subunit, such as Tyr228, Glu570 and Glu676 (Fig. 1h). The detailed interactions made by Arg241 are different among the six subunits, due to the deviation from the 6-fold symmetry of the hexamer, which again may reflect the different conformations of the arginine finger needs to adopt in the catalytic cycle (Supplementary Fig. 2b).

We carried out size-exclusion chromatography and cross-linking assays to assess the oligomerization state of the soluble region of Cagβ.

The migration of the purified Cagβ protein on gel filtration chromatography became faster at higher concentrations, suggesting concentration-dependent oligomerization (Supplementary Fig. 3a). The presence of the monomer species at lower concentrations suggests that the hexamer is not very stable, which may allow the conversion between the monomeric and hexameric states under the control of CagZ (see below). The majority of Cagβ migrated as a higher molecular weight band when cross-linked by disuccinimidyl suberate (DSS), further supporting the hexamer formation (Supplementary Fig. 3b).

## Identification of the AAD of Cagβ as the domain that binds CagZ

The domain in Cagβ responsible for the interaction with CagZ was not known. We designed a series of Cagβ constructs to express proteins for mapping the binding site for CagZ, including: Cagβ$_1$(residues 166–748), Cagβ$_2$ (residues 243–748), Cagβ$_3$ (residues 260–748), Cagβ$_4$ (residues 335–748), Cagβ$_5$ (residues 380–748), Cagβ$_6$ (residues 460–748), Cagβ$_7$ (residues 540–748), Cagβ$_8$ (residues 591–748), Cagβ$_9$ (residues 630–748) and Cagβ$_{10}$ (residues 701–748). Cagβ$_3$, Cagβ$_4$, Cagβ$_5$, and Cagβ$_6$ failed to yield soluble protein, and therefore not included in further analyses (Supplementary Table 2). Purified protein from the remaining constructs were examined for the interaction with CagZ by using isothermal titration calorimetry (ITC). The results showed that Cagβ$_1$ and Cagβ$_2$ have nearly the same affinities for CagZ, while Cagβ$_{7-10}$ did not show detectable binding (Supplementary Table 2 and Supplementary Fig. 4a). These results indicated that the region spanning residues 243–539 in Cagβ, which includes the entire AAD based on our structure of Cagβ$_1$, is necessary for the interaction with CagZ. To further narrow down the region for CagZ binding, we made three shorter constructs containing the AAD region, AAD$_1$ (residues 289–488), AAD$_2$ (residues 289–505), and AAD$_3$ (residues 299–488). Proteins from these constructs all showed high-affinity binding to CagZ, similar to Cagβ$_1$ and Cagβ$_2$, confirming that the AAD

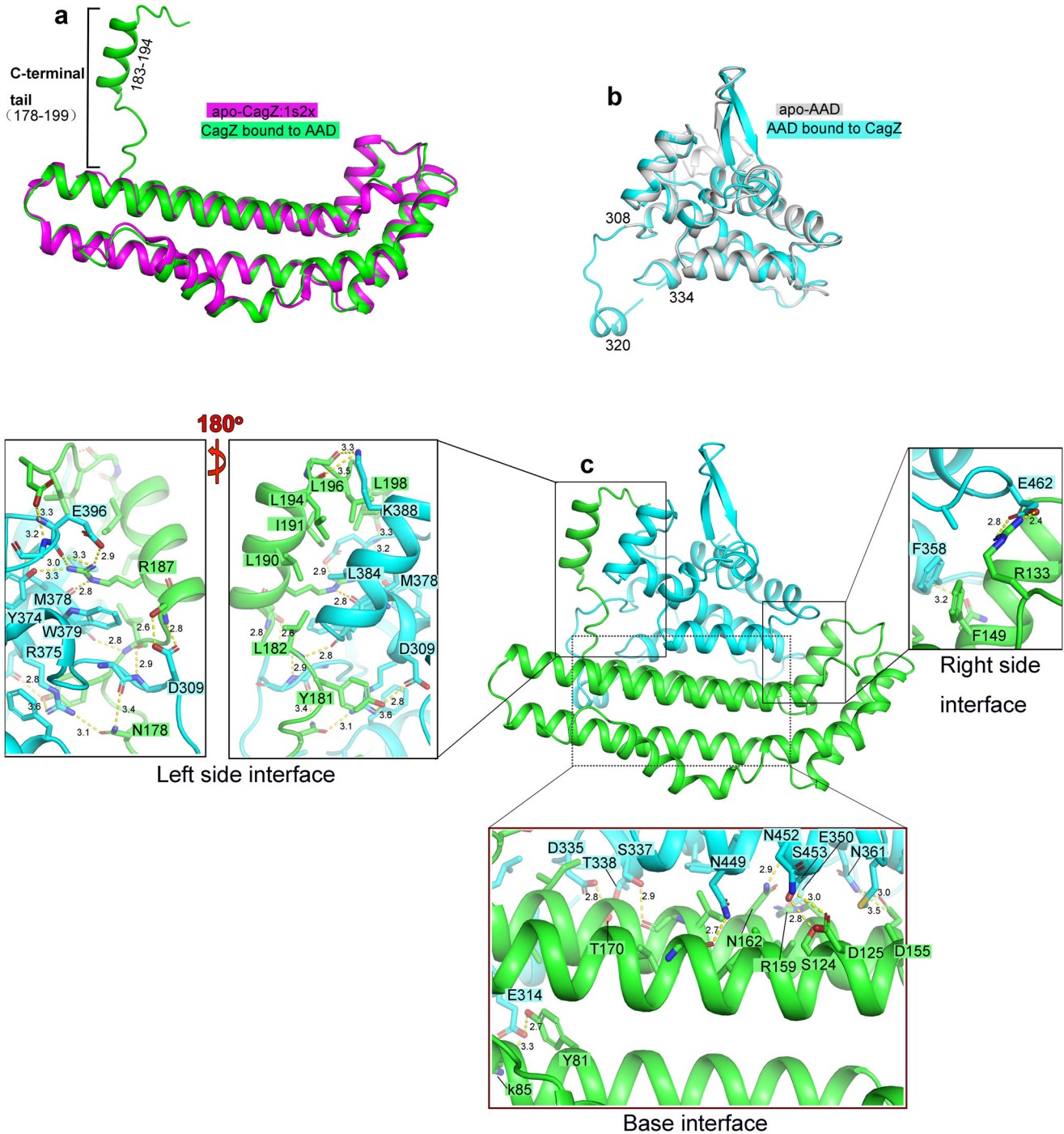

**Fig. 3 | Structure of the Cagβ-AAD₁/CagZ complex. a** Structural comparison of CagZ in the complex with its apo-state (PDB ID: 1S2X). **b** Structural comparison of Cagβ-AAD₁ in the complex with apo-AAD. **c** Structure of the Cagβ-AAD₁/CagZ complex. Details of the binding interfaces between CagZ and Cagβ-AAD are shown in expanded panels.

of Cagβ is necessary and sufficient for the interaction with CagZ (Supplementary Table 2 and Supplementary Fig. 4b).

**The structure of the Cagβ-AAD/CagZ complex**

We obtained high-quality crystals of complexes between Cagβ-AAD₁ and CagZ, which diffracted to 2.1 Å resolution (Supplementary Table 1). We solved the structure by molecular replacement using apo-CagZ (PDB ID: 1S2X) as the search model. The asymmetric unit of the crystal contains two complexes, which are essentially identical to each other (r.m.s.d. 0.11 Å). The AAD in the complex structure is very similar to that in apo-Cagβ, except for the long loop spanning residues 308 to

334 (Fig. 3b). This loop, located at the periphery of the hexameric ring, is largely disordered in the apo structure. In contrast, the middle portion of this loop (residues 307–320) turns into a short helix in the AAD/CagZ complex structure.

The structure of CagZ in the AAD/CagZ complex displays a notable difference compared with CagZ in the apo-state (PDB ID: 1S2X)[38]. While the main body of CagZ (residues 1–177) in the complex adopts the same L-shaped helical bundle structure, the previously disordered C-terminal tail (residues 178–199) becomes well-ordered. Residues 183–194 form a helix that is oriented perpendicular to the long side of the L-shaped main body, leading to an overall U-shape of

the protein (Fig. 3a). By adopting the U-shape, CagZ tightly embraces the AAD of Cagβ from three sides, resulting a large binding interface with the total buried surface area of ~2100 Å$^2$ (Fig. 3c).

The C-terminal tail of CagZ makes extensive interactions with the AAD of Cagβ (Fig. 3c, Left side interface). The middle helical segment of the tail is amphipathic in nature and packs its hydrophobic side against two helices in the AAD. L190, I191 and L194 from the tail helix in CagZ form a hydrophobic patch with M378 and L384 in the AAD. R187 in the tail helix makes a cation-pi interaction with W379. In addition, the guanidinium group of R187 form polar interactions with M378, Y374 and E396 in the AAD. The hydrophobic interface is further augmented by L196 and L198 following the tail helix of CagZ. In addition, the loop prior to the tail helix contributes both hydrophobic (Y181 and L182) and polar residues (N178) to the interface with the AAD. The bottom of the U-shape of CagZ composed of the helix bundle provides the base for the AAD to sit on, with a mixture of polar and hydrophobic residues from both proteins forming the large binding interface (Fig. 3c, Base interface). Many interactions are also made at other end of the U-shaped CagZ. For example, F149 in CagZ forms a pi-pi stacking interaction with F358 in the AAD (Fig. 3c, Right side interface). These extensive interactions provide a structure basis for the high affinity binding between CagZ and Cagβ.

To test the binding interface shown in the crystal structure of the complex, we designed mutations in both CagZ and the AAD and examined their effects on the binding between two proteins by affinity measurements with ITC. Results are summarized in Table 1, Fig. 4 and Supplementary Fig. 5. Notably, mutation of R187 in the C-terminal helix of CagZ led to a ~187-fold reduction in the binding affinity. Mutating N178 and Y181 in the loop prior to the C-terminal helix reduced the affinity by ~1.6 and ~6.2-fold, respectively. Combinations of these mutations rendered the binding undetectable (Table 1, Fig. 4a, c). These results together demonstrate the critical role the C-terminal tail of CagZ in the interaction with Cagβ. At the other end of the U-shape, disrupting the pi-pi stacking interaction by mutating F149 of CagZ or F358 led to ~4.5-fold reduction in the binding affinity, demonstrating this hydrophobic interaction is also important for the interaction (Table 1, Fig. 4b). Mutations of residues in the base of the U-shaped CagZ showed no detrimental effect on the binding affinity (Table 1 and Supplementary Fig. 5), suggesting that this part of the binding interface has a less critical role in the binding energy between CagZ and Cagβ.

To further dissect the binding interface between CagZ and the Cagβ-AAD, we designed several peptides corresponding to the C-terminal tail segment of CagZ. The ITC data showed that CagZ-24, a peptide spanning residues 176–199 in the CagZ tail, bound to the Cagβ-AAD$_1$ with Kd ~0.33 μM, approximately 3.6-fold weaker than full-length CagZ. CagZ-20 (residues 176–195) showed much weaker affinity, while further shortening of the tail abolished the binding (Fig. 4d, Table 1). These results confirm that the tail segment in CagZ makes a major contribution to the binding energy in the interaction with Cagβ.

### Binding of CagZ traps Cagβ in the monomer state

To understand how the binding of CagZ to the AAD may regulate Cagβ, we compared the AAD/CagZ complex structure with the structure of the apo-Cagβ hexamer. It is immediately clear from a structural superimposition that the binding of CagZ to the AAD is incompatible with the formation of the Cagβ hexamer (Fig. 5a, b). When bound to the AAD, the helical main body of the U-shaped CagZ sits on one side of the AAD, severely clashing with the AAD from the next subunit in the Cagβ hexamer. This observation suggests that the binding CagZ traps Cagβ in the monomeric state. Consistently, our gel filtration experiments suggested that the Cagβ$_1$/CagZ complex is smaller than the Cagβ$_1$ hexamer but larger than the monomer (Fig. 5c). To confirm this model, we used mass photometry to precisely measure the molecular weight of Cagβ$_1$ and the Cagβ$_1$/CagZ complex[48,49]. The results show that Cagβ$_1$

at 1.8 μM is mostly hexameric, with the measured molecular weight of ~408 kDa (the theoretical molecular weight of the hexamer is ~402 kDa) (Fig. 5d). At a lower concentration (0.06 μM), the monomeric species (~70 kDa) becomes dominant. The Cagβ$_1$/CagZ complex at the same concentration shows a molecular weight of ~88 kDa, consistent with a 1:1 complex. These results together support the notion that Cagβ tends to form a dynamic hexameric assembly, while the binding of CagZ traps it in the monomeric state.

### CagZ inhibits the Cagβ ATPase by trapping it in the monomeric state

As discussed above, RecA-type ATPases rely on hexamer formation for their catalytic activity because the catalytic arginine finger is provided by the neighboring subunit in the hexamer. Our structure of the Cagβ hexamer shows that no arginine residue from the neighboring subunit is present near the ATP-binding site of the NBD, suggesting R241 in the Walker A motif may fulfill the role of the arginine finger in cis (Fig. 2). Consistent with this model, our activity assays showed that mutating R241 to either alanine or lysine largely abolished the ATPase activity of Cagβ (Fig. 6a). We also tested other mutations in the Walker A (K244A) and Walker B (E551A and E551Q) motifs, which also abrogated the ATPase activity. To examine whether the ATPase activity is dependent on the hexamerization, we carried out activity assays of Cagβ$_1$ in the absence or presence of CagZ. The results show that the ATPase activity of apo-Cagβ$_1$ was strongly inhibited by CagZ (Fig. 6b). In contrast, the Y181A/R187A mutant of CagZ, which does not bind Cagβ, did not inhibit the ATPase activity (Fig. 6c and Supplementary Fig. 4a). These results together support the notion that the ATPase activity of Cagβ is dependent on the formation of the hexamer, and CagZ can keep Cagβ in the inactive monomeric state. Based on the structure, we speculate that the formation of the hexamer is required for stabilizing the Walker A motif and the conformation of the sidechain of R241, which are important for the catalytic activity.

We also tested effects on the Cagβ$_1$ activity of the isolated C-terminal tail segment of CagZ, which can bind Cagβ$_1$, albeit with reduced affinity. The results showed that neither CagZ-24 nor CagZ-20 inhibited the ATPase activity of Cagβ$_1$, even when used at over 100-fold excess concentrations (Fig. 6d). Analyses of the structure show that the tail segment is bound to the outer surface of AAD, far away from the inter-subunit interface in the Cagβ hexamer (Fig. 5a). Therefore, binding of the isolated tail segment of CagZ to the AAD has no impact on the formation of the hexamer or catalytic activity of Cagβ.

## Discussion

*H. pylori* contains several adhesins that mediate adherence to gastric epithelial cells, which triggers the assembly of T4SS for delivery of CagA to host cells[20]. For *H. pylori* to respond quickly to adherence, the bacteria may need to store the protein components of T4SS prior to assembly. In this regard, our structure of the Cagβ/CagZ complex suggests a mechanism for storing a pool of Cagβ in the monomeric state on the inner membrane (Fig. 7a, step1). Our data show that monomeric Cagβ bound by CagZ is enzymatically inactive, which prevents wasting of ATP prior to assembly of the hexameric complex competent of CagA translocation. This model is also consistent with the previous study showing that CagZ plays a role in stabilizing Cagβ, as strains carrying CagZ mutations lose the Cagβ protein[35].

Our structure and binding assays together demonstrate the C-terminal tail segment in CagZ plays a dominant role in the interaction with the AAD of Cagβ. However, the C-terminal tail of CagZ itself does not impair the hexamerization or the ATPase activity of Cagβ, because it binds to the outer surface of the AAD that is distal to the inter-subunit interface in Cagβ. We speculate that the binding surface on AAD for the CagZ C-terminal tail may also serve as the docking site for CagA. It has been shown that CagA contains a C-terminal translocation signal, which is required for its secretion by T4SS[44]. It is

**Table 1 | Dissociation constants derived from isothermal titration calorimetry**

| 'U' shaped CagZ | | Cagβ-AAD$_1$ | | K$_d$ (μM) | 1σ (μM) |
|---|---|---|---|---|---|
| **WT** | | **WT** | | **0.09** | **0.08–0.11** |
| left side interface mutants (C-terminal tail) | N178A | R375A | Left side interface mutant | 8.77 | 7.35–10.86 |
| | Y181A | R375A | | 35.71 | 22.72–83.3 |
| | S184A | WT | WT | 0.07 | 0.06–0.10 |
| | N178A | WT | | 0.15 | 0.13–0.20 |
| | Y181A | WT | | 0.56 | 0.32–2.08 |
| | R187A | WT | | 16.89 | 11.33–33.22 |
| | E199A | WT | | 0.04 | 0.03–0.07 |
| | N178A/Y181A | WT | | 1.50 | 1.18–2.05 |
| | N178A/R187A | WT | | U. D. | |
| | Y181A/R187A | WT | | U. D. | |
| | N178A/Y181A/R187A | WT | | U. D. | |
| Base interface mutants | Y81F | E314A | Base interface mutants | 0.09 | 0.08–0.11 |
| | K85A | E314A | | 0.08 | 0.06–0.10 |
| | S124A | S453A | | 0.02 | 0.01–0.03 |
| | D125A | N452A | | 0.06 | 0.04–0.12 |
| | Q150A | D356A | | 0.05 | 0.04–0.06 |
| | D155A | N361A | | 0.06 | 0.05–0.06 |
| | R159A | E350A | | 0.04 | 0.03–0.06 |
| | N162A | N365A | | 0.04 | 0.03–0.11 |
| | T170A | D335A | | 0.09 | 0.06–0.24 |
| | T170A | T338A | | 0.09 | 0.06–0.24 |
| Right side interface mutants | R133A | E462A | Right side interface mutants | 0.02 | 0.01–0.03 |
| | F149A | F358A | | 0.41 | 0.28–0.75 |
| WT | WT | D309A | Right side interface mutants | 0.07 | 0.06–0.09 |
| | WT | Y374F | | 0.02 | 0.01–0.03 |
| | WT | R375A | | 1.46 | 1.34–1.60 |
| | WT | D376A | | Precipitated | |
| | WT | K388A | | 0.04 | 0.03–0.06 |
| | WT | E396A | | 0.04 | 0.03–0.08 |
| | WT | N449A | Base interface mutant | 0.02 | 0.01–0.03 |
| C-terminal peptides | CagZ-24 | WT | | 0.33 | 0.29–0.38 |
| | CagZ-20 | WT | | 42.02 | 30.97–65.32 |
| | CagZ-24 | WT | | U. D. | |
| | CagZ-12a | WT | | U. D. | |
| | CagZ-12b | WT | | U. D. | |
| | CagZ-10 | WT | | U. D. | |
| | CagZ-7 | WT | | U. D. | |

U.D. binding undetectable.

conceivable that this C-terminal translocation signal in CagA also bind to the outer surface on the AAD of Cagβ. This binding would place CagA near the entrance of the central channel of the Cagβ hexamer, ideally suited for initiating the translocation. It may also compete the binding site with the C-terminal tail of CagZ, driving CagZ dissociation and thereby promoting the assembly of the Cagβ hexamer (Fig. 7a, step 2). On the other hand, it remains possible that other factors may trigger the dissociation of CagZ and allow the hexamer formation of Cagβ in the absence of CagA.

Our structure of the Cagβ hexamer shows a narrow central channel that can accommodate unfolded polypeptides but not large, folded domains. CagA is a ~130 kDa protein with multiple folded domains that have dimensions much larger than the diameter of the central channel of the Cagβ hexamer. These analyses therefore provide a structural basis for the observations that CagA must unfold during the translocation process by previous studies[50]. Translocation of the unfolded polypeptide chain of CagA in the central channel of the Cagβ hexamer may use the same step-by-step mechanism as that of the p97 unfoldase[43]. The p97-mediated unfolding and translocation process is directional, where the N-terminal end of the substrate enters the central pore in the p97 hexamer first. p97 then uses the energy from ATP hydrolysis to pull the substrate through the central pore in a stepwise manner. Analogously, we speculate that CagA, with the C-terminal captured by the AAD of Cagβ, places its N-terminal end to the central channel of the Cagβ hexamer (Fig. 7a, step 3). Pulling of the N-terminal end of CagA by ATP hydrolysis of Cagβ leads to unfolding and translocation of CagA.

Several structures of the intact T4SS assembly of *H. pylori* have been solved by single-particle cryo-electron microscopy (SP-cryo-EM) and cryo-electron tomography (cryo-ET)[51,52]. These structures reached high resolution for the outer membrane complex of Cag T4SS, whereas the inner membrane complex was not well resolved. More recently, a SP-cryo-EM structure of the T4SS encoded by the R388 plasmid reached near atomic resolution for most parts of the complex and

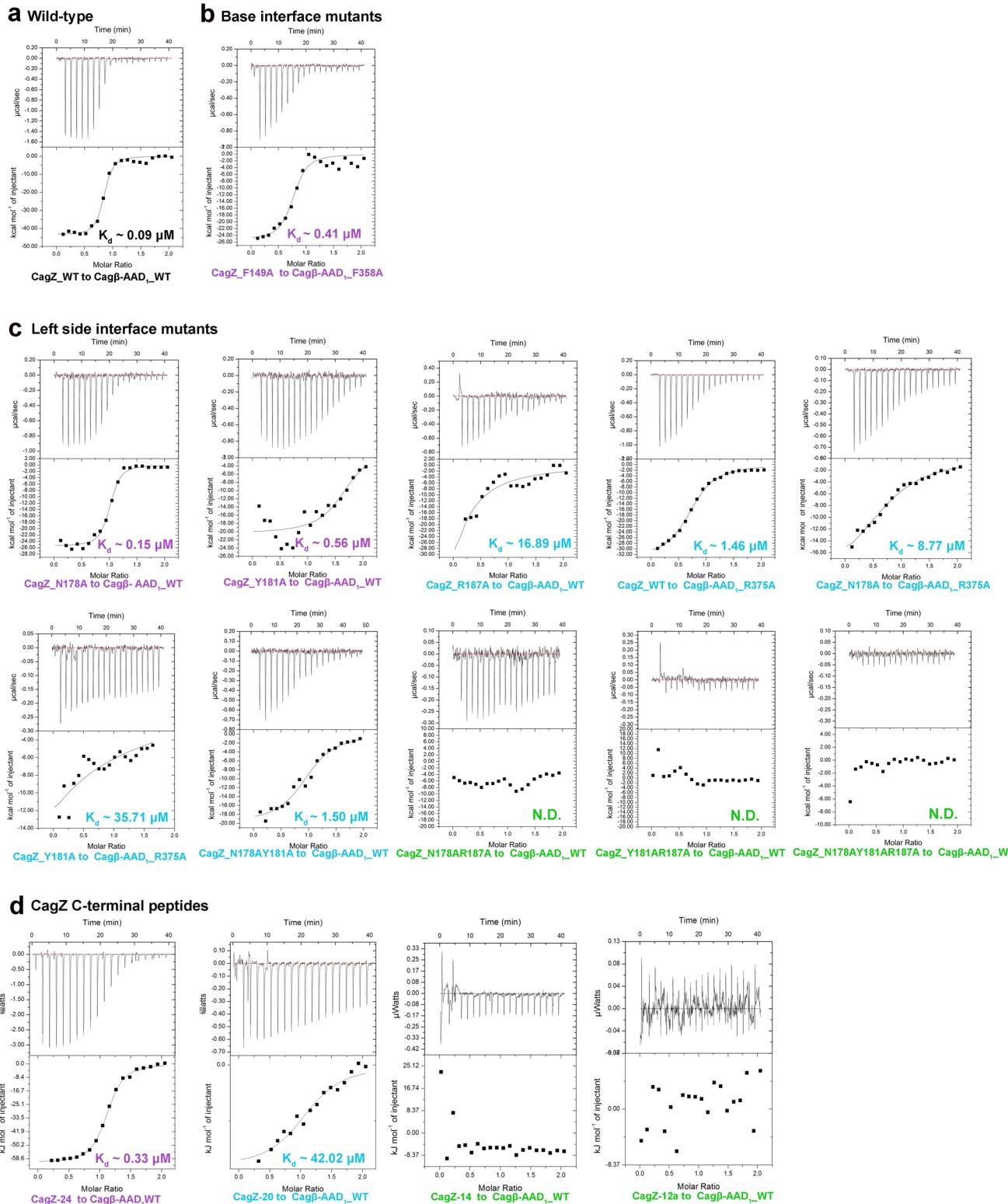

**Fig. 4 | Mutational analyses of the binding interface between CagZ and Cagβ-AAD₁ using ITC.** Different colors indicating the different levels of affinity compared with the wild type (Black, nearly the same affinity with the wild type; violet, 2–10 fold weaker; blue, 50-1000 fold reduced; green, binding undetectable). **a** high affinity binding between wild type CagZ and Cagβ-AAD₁. **b** The F149A mutant of CagZ and the F358A mutant of Cagβ-AAD₁ have significantly reduced binding affinity. **c** The effects of left side interface mutants on the binding affinity. **d** Binding affinities of various C-terminal tail peptides of CagZ for Cagβ-AAD₁.

confirmed the hexameric arrangement of the inner membrane complex, but the TrwB/VirD4 protein is missing in the structure, suggesting that it is not stably associated[53]. Nevertheless, the density map from the in situ cryo-ET study of the *H. pylori* Cag T4SS clearly showed a distinct 3-ring architecture of the inner membrane complex with 6-fold symmetry[52]. Cag T4SS from a strain with Cagβ knocked-out lost density for the outer ring as well as the membrane-distal part of the inner ring of the inner membrane complex, suggesting that these parts are

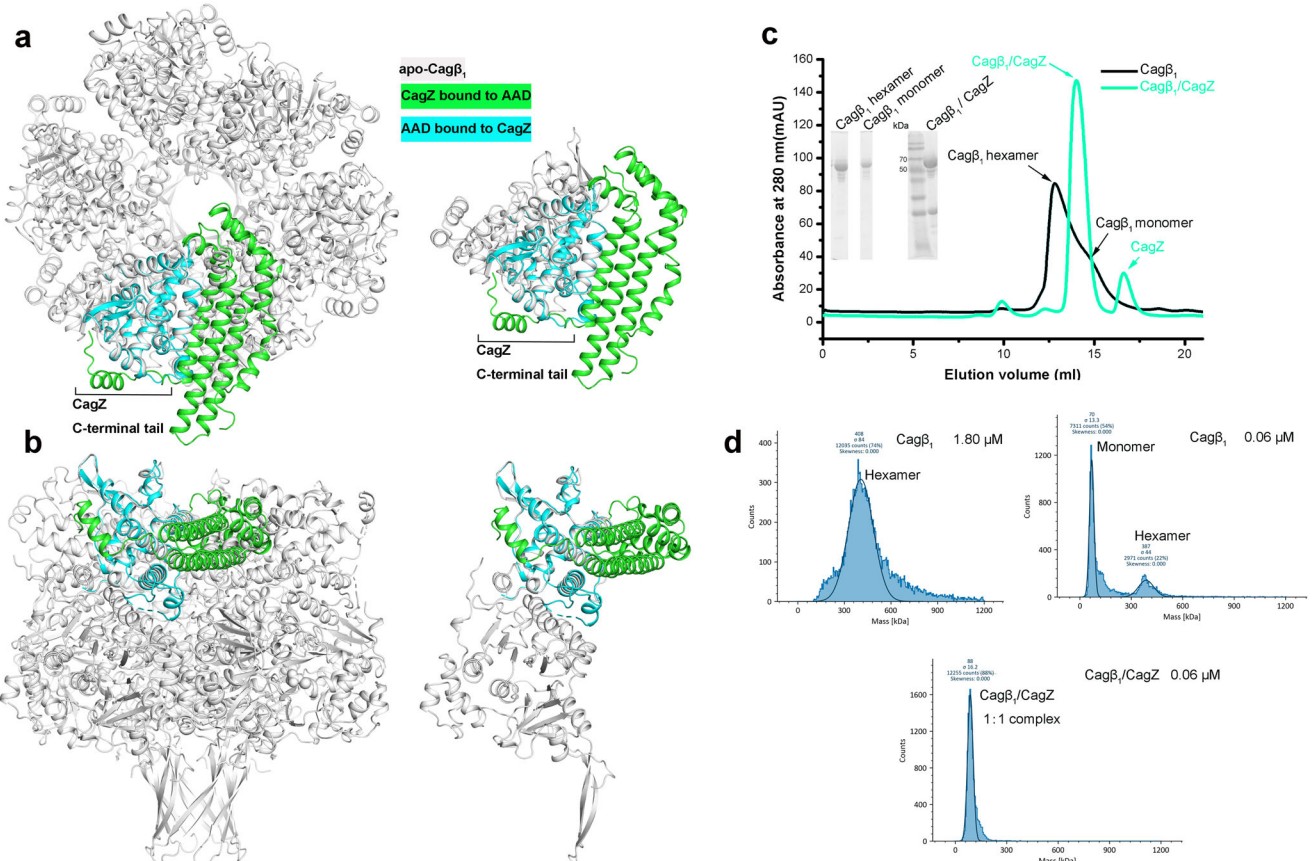

**Fig. 5 | CagZ traps Cagβ in the monomeric state. a, b** Structure superimposition of the Cagβ-AAD$_1$/CagZ complex with the apo-Cagβ hexamer or one apo-Cagβ protomer. CagZ bound to one Cagβ protomer clashes severely with the neighboring Cagβ protomer, supporting the notion that this binding is not compatible with the Cagβ hexameric assembly. **c** Size exclusion chromatography analyses of purified Cagβ$_1$ (75 μM, black) and the Cagβ$_1$/CagZ complex (110 μM, green). The inset demonstrates the purity of proteins used in experiments (Uncropped SDS-PAGE is provided in the Source Data file). **d** Mass distribution of Cagβ$_1$ and the

Cagβ$_1$/CagZ complex measured by mass photometry. Mass histogram (blue) and corresponding Gaussian fit model (black) were shown here. Approximately 74% Cagβ$_1$ ($n = 12035$ counts) at 1.8 μM is hexameric, with the measured molecular weight of ~408 kDa (the theoretical molecular weight of the hexamer is ~402 kDa). At a lower concentration (0.06 μM), ~54% ($n = 7311$ counts) of the protein is monomeric (~70 kDa) and ~22% ($n = 2971$ counts) is hexameric. The Cagβ$_1$/CagZ complex at 0.06 μM shows ~88% ($n = 12255$ counts) of the protein are in a molecular weight of 88 kDa, consistent with a 1:1 complex.

constituted of Cagβ or dependent on Cagβ for assembly. We docked our hexamer structure of soluble Cagβ into the cryo-ET map and found it fits the cone-shaped membrane-distal part of the inner ring very well (Fig. 7b). It has been proposed that the cryo-ET structure represents a quiescent state of the Cag T4SS, because CagA is absent[52]. It is possible that the Cagβ hexamer in this state is subjected to additional regulatory mechanisms that supresses the ATPase activity, while binding CagA releases this inhibition and triggers the ATPase-driven translocation process. The density in the outer ring may contain monomeric Cagβ and/or additional unknown T4SS components that dependent on Cagβ for docking to the inner membrane complex. Based on this docking model and analyses above, we propose a model of the regulation of Cagβ in Cag T4SS of *H. pylori* (Fig. 7a). Due to the low resolution of the cryo-ET map, it is not possible to discern how Cagβ interacts with other components of the inner membrane complex and how the soluble domain of Cagβ is attached to the membrane by its transmembrane helix. Further structural analyses are required for addressing these questions of great mechanistic significance.

## Methods
### Bacterial strains, transformation and plasmid constructs
*E. Coli* strain XL10-gold (Invitrogen) was grown on Luria-Bertani (LB) agar plate or in LB liquid medium supplemented with ampicillin (100 μg/mL) and/or Kanamycin (50 μg/mL). The *H. pylori* 26695 genome was extracted by following the protocol of Bacteria Genomic DNA

Kit (Zoman ZP301). The genome DNA was used as template for cloning CagZ (residues 2–199) and all constructs of Cagβ (Supplementary Table 3).

### Protein expression and purification
All Cagβ, CagZ fragments used in the study were cloned into the pET-32a vector in separate expression, which encodes a N-terminal His$_6$-tag and TrxA-tag (thioredoxin protein) followed by a Tobacco Etch Virus (TEV) protease cleavage site. For the co-expression, CagZ was cloned to pET-28a which only contains a N-terminal His$_6$-tag. Plasmids were transformed into the *E. Coli* strain BL21 star (DE3) (Invitrogen) for protein expression. Cells were grown at 37 °C in LB medium, and induced by 0.3 mM IPTG when the optical density of the culture reached ~0.8. Cells were further cultured overnight at 16 °C. Cells were harvested by centrifugation and resuspended in the Ni-NTA buffer (50 mM Tris pH 8.0, 500 mM NaCl, 10% glycerol) supplemented with 0.1 mM EDTA, 20 mM Imidazole, 0.1% Tween-20, and 2 mM β-mercaptoethanol. Cells were lysed by sonication and clarified lysate were loaded onto Ni-NTA columns. Captured proteins were washed with the same buffer extensively and eluted with the elution buffer containing 300 mM imidazole. The His$_6$-tag was cleaved by the TEV protease, and removed by a second around of Ni-NTA purification. The proteins were further purified by size exclusion chromatography (Superdex™ 200 Increase 10/300, GE Life Sciences) in a buffer containing 25 mM Tris pH 8.0, 150 mM NaCl, 5% glycerol. For expression of

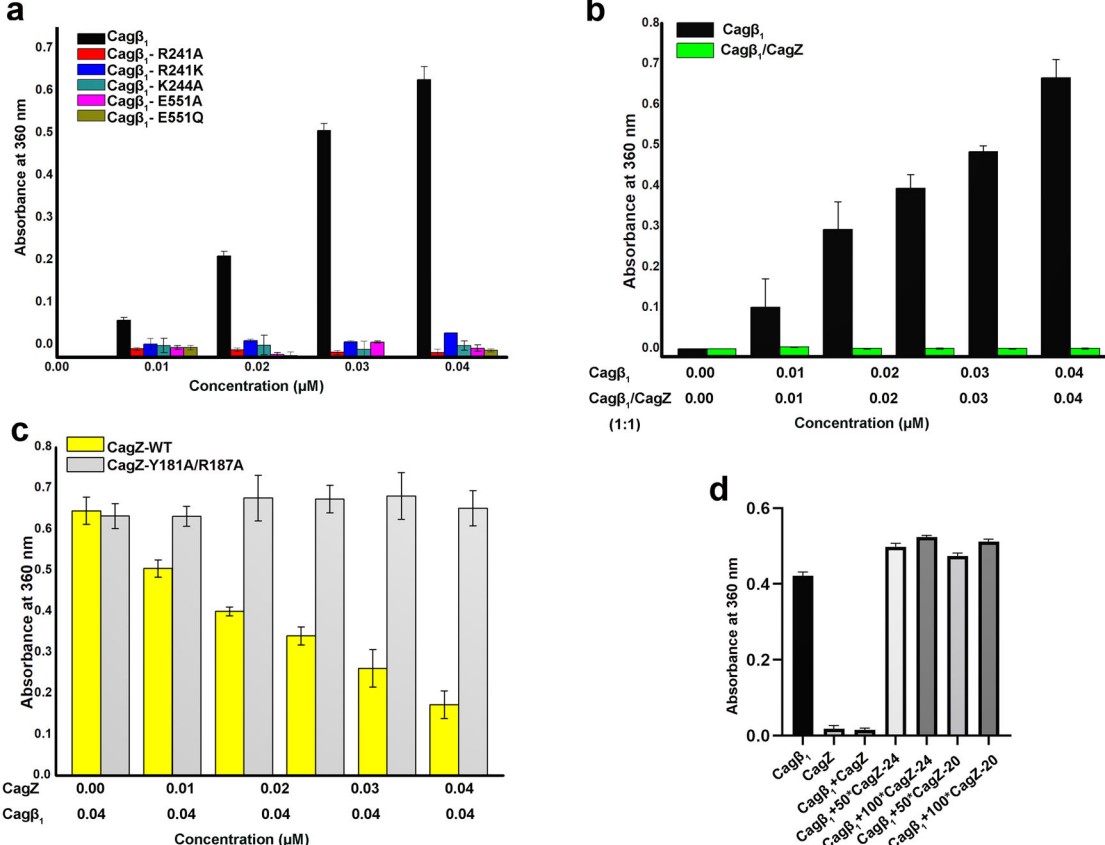

**Fig. 6 | ATPase activity assays of Cagβ₁. a** ATPase activity of apo-Cagβ₁ wild type and ATP binding site mutants measured by a photometric assay (see details in methods). **b** CagZ inhibits the ATPase activity of Cagβ₁. **c** the Y181A/R187A mutations of CagZ, which disrupt the interaction with Cagβ₁, abolish the inhibition on the Cagβ₁ ATPase activity. **d** C-terminal peptides of CagZ do not inhibit the ATPase activity of Cagβ₁. Values are averages from three replicates. Error bars indicate the standard deviation from three biological repeats ($n$ = 3 for all groups). Source data are provided as a Source Data file.

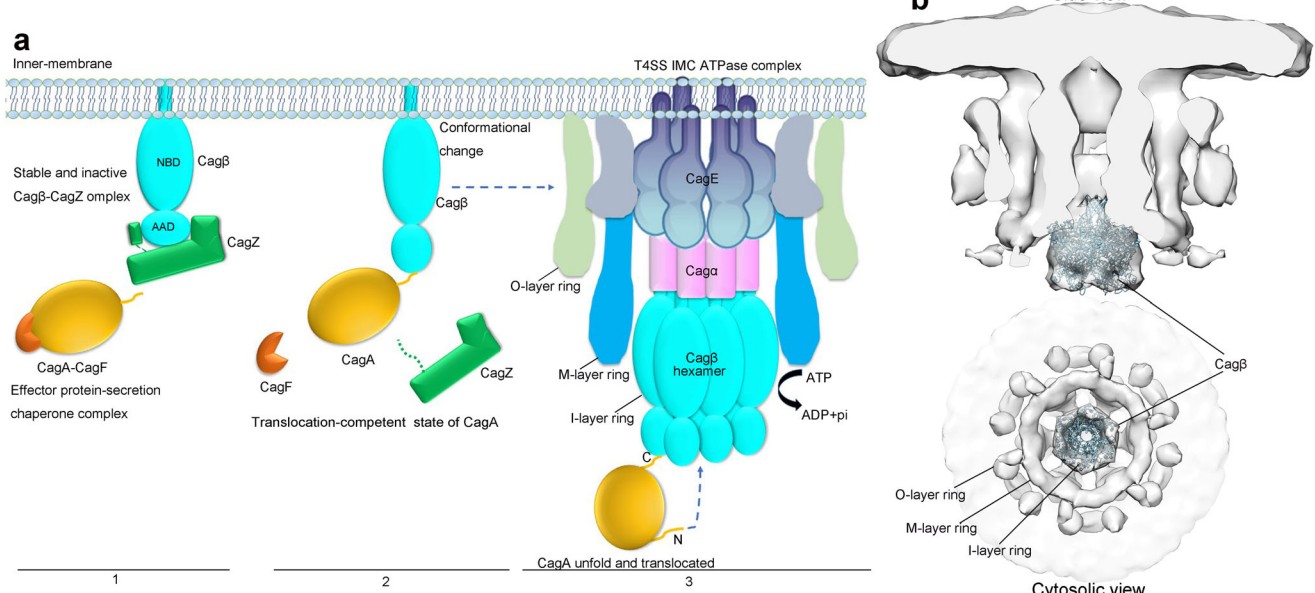

**Fig. 7 | A model for CagA recruitment by Cagβ of the Cag T4SS. a** 1, CagA forms an effector-secretion chaperone complex with CagF[36,63] near the inner membrane, where Cagβ is trapped in the inactive monomeric state by CagZ. 2, Binding of CagA triggers the dissociation of Cagβ from CagZ. 3, Cagβ forms the hexamer and docks to the inner membrane complex of T4SS, ready for translocating CagA through the central channel. **b** Docking model of the Cagβ hexamer to the in situ Cryo-ET density map (EMD ID: 0634).

the complex between CagZ and Cagβ1 or Cagβ2, we co-transformed constructs of Cagβ-pET-32a and CagZ-pET-28a into BL21 star (DE3) cells. Complex proteins were purified by following same procedures as above mentioned, except that all buffers contained no glycerol and the concentration of NaCl was reduced to 100 mM NaCl. To obtain CagZ in complex with AAD1, AAD2, or AAD3, CagZ was mixed with the Cagβ-AAD proteins at a molar ratio of 1.1:1, and incubated overnight at 4 °C. The protein complexes were purified by the same size exclusion chromatography procedure as above.

Mutations of Cagβ and CagZ were generated using a QuikChange site-directed mutagenesis kit (Stratagene) and confirmed via DNA sequencing. Mutant proteins were expressed and purified with the same procedure as for the wild type described above, but without TEV protease treatment.

### Crystallization, data collection, structure determination

Cagβ1 and Cagβ2 at 6 mg/ml were subjected to crystallization screens by the sitting-drop vapor diffusion method. High-quality crystals of Cagβ1 appeared in the condition containing 35% PEG400, 0.2 M Li2SO4, 0.1 M Tris 8.5 at 285 K. Crystals were flash-frozen in liquid nitrogen without additional cryoprotectant. The Cagβ-AAD1/CagZ complexes were concentrated to 20 mg/ml for crystallization. Crystals of the Cagβ-AAD1/CagZ complex were obtained in 20% PEG8K, 0.2 M sodium chloride, 0.05 M CAPSO, pH 10.5. Crystals were flash-frozen in liquid nitrogen in the crystallization buffer supplemented with 20% glycerol as the cryoprotectant.

The diffraction datasets for apo-Cagβ1 and the Cagβ-AAD1/CagZ complex were collected at 100 K on the BL19U1 beamline at the Shanghai Synchrotron Radiation Facility (SSRF) to resolution of 2.8 and 2.1 Å, respectively. Data were processed with HKL2000[54]. Structures were determined by the molecular replacement using the PHASER module in the Phenix package[55,56]. The search model for the Cagβ1 structure was generated using Alphafold as implemented in ColabFold[39,40]. Phaser found all six protomers in the asymmetric unit, leading to an electron density map of sufficient quality for manual model building. The structure of apo-CagZ (PDB ID: 1S2X) was used as the search model for the Cagβ-AAD1/CagZ complex. The search found two protomers of CagZ in the asymmetric unit and generated a map that showed density for the AAD of Cagβ. Iterative manual model building and refinement were performed with the COOT[57] and PHENIX[58], respectively. Structure validation was carried out using the Molprobity module in Phenix[59]. Structure figures were rendered in Coot, Pymol (The PyMOL Molecular Graphics System, Schrödinger) or chimera1.16[60]. Sequence alignment was rendered with ESPript 3[61]. Buried surface area is calculated by using the PDBe PISA server[62]. Data collection and model refinement statistics are summarized in Supplementary Table 1.

### Peptide synthesis

Peptides corresponding to the CagZ C-terminal tail were synthesized by Genscript. Following peptides were generated: CagZ-24 (SKNKTYLTSLERAKLITQLKLNLE), CagZ-20 (SKNKTYLTSLER-AKLITQLK), CagZ-14 (SKNKTYLTSLERAK), CagZ-12a (SKNKTYLTSLER), CagZ-12b (NKTYLTSLERAK), CagZ-10 (NKTYLTSLER) and CagZ-7 (YLTSLER).

### Isothermal titration calorimetry

Measurements were conducted using an ITC-200 microcalorimeter (MicroCal) at 25 °C. All samples were dialyzed into the buffer (20 mM Tris pH 8.0, 150 mM NaCl) prior to ITC experiments. The sample cell (300 μl) was filled with the Cagβ proteins (20 μM), and the injection syringe (40 μl) was filled with the CagZ proteins (200 μM). The experimental parameters were as follows: 20 injections, 2 μl and 1 s per injection, with an interval of 150 s and stirring speed of 1000 rpm. The data were analyzed and fitted using the Origin Pro8 software suite.

### Chemical cross-linking assay

For cross-linking experiments, the Cagβ1 protein was exchanged into the buffer containing 20 mM Hepes pH 7.0, 150 mM NaCl. The cross-linking agent disuccinimidyl suberate (DSS, Pierce™) at 50 mM in DMSO was added to different final concentrations (0.1–2.0 mM) to Cagβ1 at 1.5 mg/ml. Samples were mixed and incubated for 30 min at room temperature. Reactions were quenched by adding Tris pH 7.5 to a final concentration of 50 mM. Samples were analyzed by 15% SDS-PAGE.

### Mass photometry

The mass photometry experiments were performed following the standard procedure[49]. Briefly, the bovine serum albumin (BSA) protein was used as the protein standard to calibrate the OneMP instrument (Refeyn). In general, 16.8ul buffer is added to find the focus, which is followed by adding 1.8ul protein sample at various concentrations. Mass photometry signals were collected with the AcquireMP software for 60 s to detect at least 10000 individual molecules. Raw data were processed by using the DiscoverMP software, plotted as histograms and fitted to obtain the molar mass distribution.

### ATPase assay

The ATPase activity of Cagβ was measured by following the release of inorganic phosphate upon ATP hydrolysis by using the EnzChek™ Pyrophosphate Assay Kit (E-6645). ATPase reactions were initiated by mixing Cagβ1 or the Cagβ1/CagZ complex at various concentrations with the reaction buffer containing 50 mM Tris pH 7.5, 0.4 mM ATP and 1 mM MgCl2. After incubation for 60 min at room temperature, the released inorganic phosphate was measured by the enzyme-coupled assay in which phosphate and 2-amino-6-mercapto-7-methylpurine ribonucleoside (MESG) were converted by purine nucleoside phosphorylase (PNP) into ribose-1-phosphate and a guanine base, leading to an increase in absorbance at 360 nm that could be detected photometrically.

### Reporting summary

Further information on research design is available in the Nature Portfolio Reporting Summary linked to this article.

## Data availability

The atomic coordinates and structure factors in this study have been deposited in the Protein Data Bank database under accession code 8DOL (Cagβ1), 6JHO (Cagβ-AAD1/CagZ), respectively. For reference structures used in this paper, 1E9R (TrwB) and 1S2X (CagZ). All the relevant data are available from the authors. Source data are provided with this paper.

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

## Acknowledgements

Authors thank staffs from BL17U1/BL18U/BL19U beamline of National Facility for Protein Science in Shanghai (NFPS) at Shanghai Synchrotron Radiation Facility (SSRF) for assistance during data collection. This work was supported by the Key Project of Fujian Province (2017N0031), the STS project of Chinese Academy of Sciences and Fujian Province (2016T3041), the National Nature Science Foundation of China (31470741), Special Funds of the Central Government Guiding Local Science and Technology Development (2017L3009), and National Thousand Talents Program of China. X.Z. is supported by the Welch Foundation (I-1702).

## Author contributions

X.W. and Y.W. designed research; X.W., H.Z., W.Y, L.S and J.Y. performed research; X.W., Y.Z, X.Z. and Y.W. analyzed data; X.W., Y.Z., X.Z. and Y.W. wrote the paper; X.W.,Y.Z., M.J, Qin W., QianC.W., X.Y., X.Z., and Y.W. revised the manuscript.

## Competing interests

The authors declare no competing interests.
