## [Peer Review File · Nature Communications]

Mechanism of regulation of the *Helicobacter pylori* Cag β ATPase by CagZREVIEWER COMMENTS

Reviewer #1 (Remarks to the Author):

In this manuscript, Wu et al. describe crystal structures of the *Helicobacter pylori* ATPase Cagbeta and a complex formed between a fragment of Cagbeta and CagZ, a protein that interacts with Cagbeta and regulates the translocation of the oncoprotein CagA through the type 4 secretion system into gastric epithelial cells where it dysregulates cell signaling in numerous ways that can lead to peptic ulcer diseases and gastric cancer. The authors show that full-length Cagbeta forms a hexameric ring structure and that the Cagbeta fragment bound to CagZ forms a 1:1 complex that is incompatible with hexameric formation. Furthermore, they performed activity assays to show that the ATPase activity of Cagbeta is inhibited in a dose-dependent manner by CagZ, but not a mutant of CagZ that does not bind to Cagbeta. The data is quite compelling and its presentation is generally fine. A few major points of concern follow:

1. Much of the authors' conclusions rest on the structure of the Cagbeta fragment-CagZ complex and the size exclusion chromatography analysis of full-length Cagbeta in the presence and absence of CagZ. In this crystal structure, there are two notable features: first, the Cagbeta is a fragment and, thus, not the natural full-length Cagbeta protein but an arbitrarily truncated portion; second, the C-terminal tail of CagZ in this structure adopts a conformation that is different from that of the apo CagZ structure and, moreover, makes numerous interactions with the Cagbeta fragment, which also undergoes conformational changes relative to its apo structure. Whether these conformational changes in both proteins and the resulting intermolecular interactions are caused by crystallization or not is unclear. Some solution biophysics could help here. There are numerous options, but performing a hydrogen-deuterium exchange-mass spectrometry experiment with the proteins in the crystal structure, as well as with full-length Cagbeta, would show both the interfaces gained by the interactions of these proteins and the interfaces lost in the proposed CagZ-mediated transition of full-length Cagbeta from hexameric to monomeric forms. These HDX-MS experiments, as well as additional analytical ultracentrifuge experiments, would also greatly corroborate the SEC data suggesting that CagZ transitions Cagbeta from hexamer to monomer (SEC is not the highest fidelity readout of oligomeric state).

2. The authors' structure of full-length Cagbeta shows that its Walker A motifs responsible for ATP binding and hydrolysis are formed within a single Cagbeta protomer (in "cis"), rather than through the participation of residues from neighboring protomers (in "trans") that is common to many Cagbeta homologs. This begs the question: why would CagZ-mediated trapping of a monomeric Cagbeta state inhibit ATP hydrolysis if a monomer still contains a whole Walker A motif? That is, what are the CagZ-driven changes, conformational or otherwise, in the Cagbeta Walker A motif that explain the related dysfunction of the ATPase activity? Again, HDX-MS could indicate conformational and/dynamic changes in the Cagbeta Walker A motifs induced by CagZ.

Some minor comments follow:

1. There are a number of instances in which the order of figures and/or figure panels appear out of order with the text (e.g., figure 2 is described in text after the early panels of figure 1 but before the latter panels of figure 1; figure 7b is described before 7a).

2. Line 204: "biochemical assays" is vague.

3. Line 208: "superstable" - is this a technical term?

Reviewer #2 (Remarks to the Author):

This manuscript reports an exciting advance in the T4SS field concerning the structural

definition of the Cagb coupling protein and effects of the CagZ adaptor on Cagb structure and catalytic activity. The Cagb structure is only the second X-ray structure of the T4CP components of T4SSs, which allowed for detailed comparisons between Cagb and the current structural prototype for T4CPs, TrwB. The T4CPs are essential for translocation of DNA and protein substrates through most T4SSs, therefore, it is critical to know their architectures and effects of modulatory proteins such as CagZ. The apostructure of CagZ was previously reported, and here the authors show that CagZ undergoes a profound conformational change when bound to the all-alpha-domain (AAD) of Cagb. Previous studies have shown that T4CP AADs are important receptor domains for secretion substrates, and here the authors propose that CagZ coordinates binding of the secretion substrate CagA with the Cagb T4CP for translocation through the Cag T4SS. CagZ locks Cagb into the monomeric state and also blocks its ATPase activity, suggesting that CagZ binding renders Cagb inactive. Their findings prompt a model in which the binding of CagA to the CagZ-Cagb complex results in dissociation of CagZ, which in turn enables Cagb hexamer formation and catalytic activity as a prerequisite for Cagb-mediated unfolding and translocation of CagA. Overall, the work is well-described and appears technically sound although this reviewer is not an X-ray crystallographer. It represents a major advance in the field insofar as there is no prior information describing in atomic detail the structural effects of an adaptor on the nucleotide binding domain of a T4CP. There are a number of grammatical flaws that can be easily rectified. As mentioned below, the authors should consider their model in the context of the recent CryoET structure of the Cag machine in the *H. pylori* envelope, which shows densities potentially corresponding to the Cagb hexamer at the base of the central channel. Other issues are minor.

1. Pg. 5, L. 139 Walker. Fix other grammatical errors throughout.

2. Pg. 6. L. 150. This isn't evident from the presented structures, which isn't that important, but if the authors want to highlight this point, a different color scheme should be used for the C-terminal domain.

3. Pg. 6. L. 155. Not evident since there's no side-by-side comparison in this figure. There is a comparison in the suppl figure 1a,b, but this also does not reveal the size difference.

4. Pg. 7. L. 226. Since no information can be gained for residues 260-460, it is possible that these residues also bind CagZ even though the AAD also does.

5. Fig. 4 requires a fuller description in the legend about the data shown.

6. Fig. 7. and Discussion. The model presented in Fig. 7 should be referred to early in the Discussion, even within the first paragraph. The model presents a flow of step-wise binding reactions that can be referred to throughout the Discussion.

7. The model is quite interesting, but needs to be reconciled with a couple of observations. First, a *deltacagZ* mutant lacks detectable CagA and, second, the CryoET structure shows densities consistent with a Cagb hexamer at the base of the inner ring. Relating to the *deltacagZ* phenotype, it isn't clear why *cagZ* binding to the Cagb monomer destabilizes CagA. Relating to the Cagb hexamer-IMC architecture detected by CryoET, this is assumed to represent the Cag machine in its quiescent state. If this is the case, it isn't obvious why Cagb assembles as the catalytically hexamer in the quiescent machine. Couldn't another model simply be that Cagb assembles as a hexamer but when docked onto the Cag machine it is catalytically inactive? CagZ could function as an adaptor by binding CagA and recruiting it to the AAD. Upon CagA-CagZ-Cagb binding, the CagZ - Cagb interaction induces conformational changes in the Cagb hexamer that do not necessarily induce monomerization but instead induces structural changes necessary for docking of CagA, an event that stimulates Cagb catalytic activity and, accordingly, CagA unfolding and translocation. This model is simpler and more consistent with results of the CryoET studies and CagZ-CagA biochemical studies.

Regardless of whether the authors like this latter model, they need to do a better job reconciling their findings with the observations mentioned above.

Reviewer #3 (Remarks to the Author):

In the manuscript entitled "Mechanism of regulation of the *Helicobacter pylori* Cag β ATPase by CagZ" Wu et al. detail the structural analysis of the *H. pylori* protein known as Cag β , an ATPase associated with the *H. pylori* Cag type four secretion system (T4SS). The function of Cag β is to aid in the transference of the oncogenic protein CagA, into host cells. In this manuscript, the authors detail two structures of Cag β , one in a hexameric assembly and one in the monomeric form bound to the accessory protein CagZ. The analysis described here highlights a regulatory mechanism for CagZ that suggests it prevents premature oligomerization of Cag β . The authors provide evidence for this mechanism using isothermal titration calorimetry, ATPase activity assays, and size exclusion chromatography. Although the observation is noteworthy and the presented mechanism interesting, it should be noted that several critical pieces of data are missing from the manuscript. Coupled with a severe lack of clarity, it is suggested that this manuscript is not suitable for publication. Noted below are several issues that have been raised for this manuscript.

Major Issues

1. The authors make several claims about the binding site of ATP in Cag β but do not present a crystal structure with ATP bound. In fact, most of the analysis concerning the ATP binding site is dependent on the placement of a SO₄²⁻ ion in the active site. Without a structure of Cag β bound to ATP, these comments are speculative. It is suggested that the authors work towards a co-crystal structure of Cag β in complex with ATP, or a non-hydrolysable analog.
2. The evaluation of the ATP binding site is interesting as the authors highlight the lack of an arginine finger between adjacent protomers in the hexameric assembly and suggest that another residue (Arg241) takes its place in cis. The authors have clearly established an ATP hydrolysis assay, and as such, it would greatly benefit their analysis to test active site mutants for ATPase activity.
3. Crystallographic interactions are indicated in several figures throughout the manuscript, but no distances are given. At present, it is not clear to the reader how these interfaces have been determined. The authors should define this in the manuscript and label these interactions in their figures.
4. The authors have submitted several preliminary PDB validation reports for the Cag β structure that are all clearly marked "not for manuscript review". The authors should provide a full, official validation report with their submission.
5. The authors present derived K_d values for a library of CagZ peptides binding to Cag β -AAD but do not present any data for these experiments. The data for these experiments must be included in the manuscript.
6. The authors present binding data for CagZ to Cag β (Extended Data Figure 5a), but the data appear to be very noisy. At present, it is hard to understand how the affinity for this interaction was determined and to what degree the readers can be confident in the reported K_d. The authors should provide additional experimental support for the determined K_d or remove this analysis from the manuscript.
7. The authors present size exclusion chromatography data describing the oligomerization state of Cag β at high concentrations. As presented, it is unclear if the shift in retention volume is consistent with an ~6-fold shift in molecular weight. The

authors should provide additional validation to support this claim.

8. The authors provide crosslinking data to demonstrate the oligomerization of Cag β at high concentrations. As presented, the data are unclear, and it seems that most of the protein is aggregating when cross-linked. The authors should present a calibrated molecular size marker to indicate the molecular weight of the species that is formed in the presence of DSS.

Minor Issues

1. Line 50: 'strains of *H. pylori* use their T4SS' should directly reference the Cag T4SS as *H. pylori* contains up to four T4SSs in total.

2. Line 84 references the T4SS in *H. pylori*; again, the authors should specify that the T4SS referenced is the Cag T4SS.

3. Line 107 is confusing as written. It is assumed that the authors mean that the determined structure consists of residues 166-748, though it implies that the transmembrane domain is formed by residues 166-748. The authors should also reference Figure 1a here.

4. Extended data Figure 1a – the gray coloring of TrwB is difficult to see as presented. It would be best to change the color of TrwB to better illustrate the observed differences.

5. It is suggested that the plot in Figure 1c be rotated.

6. In line 123 and Figure 1b; the figure does not clearly illustrate the authors' comment concerning the fold of the NBD. The figure or the text should be reworked to better illustrate this point.

7. In Figure 2, residues forming the Walker A and/or B motifs should be noted in the figure to better orient the reader.

8. In line 127, the Walker A motif is referred to as 'Walk A'. This happens several times in the same paragraph. These need to be corrected.

9. In line 137, Walker A should be capitalized.

10. In line 130, the authors describe density in the nucleotide binding site but show only the model. The referenced density should be shown in the figure.

11. In lines 134-136 the authors argue that Arg241 of Cag β serves the same function as Arg375 of TrwB. The authors should show an overlay of the two active sites to illustrate this point.

12. In lines 142-143 the authors describe residues 631-658 as forming 'the base layer of the 3-layered Cag β '. This is confusing. Are the authors referring to the AAD, NBD, and B-barrel? It is also not clear what 'base layer' means in this context.

13. It would be useful to the readers if the authors showed a comparison between the β -strands of Cag β with those observed in TrwB.

14. In Figure 1d the authors show a view of all orientations of Cag β that are observed in the hexameric form. The authors should elaborate on how the AAD changes from one protomer to another. Does it rotate or translate and to what degree?

15. On line 157 the authors reference the larger structure of Cag β compared to TrwB, but they do not illustrate this in the referenced figure (Figure 1e).

16. On line 159 the authors reference the ATP binding site at the inter-subunit interface and reference Figure 1h. It is not clear where the proposed ATP binding site is in this figure.

17. In Figure 1g the authors intended to show the lack of interactions between adjacent AAD domains within the hexameric structure. This is not clearly shown. In fact, it looks like the authors even highlight a single interaction. It would help to demonstrate the distances observed between residues.

18. In Figure 1d the authors reference the positions of the AADs in the hexamer and note that protomers A, B, and C have different orientations than D, E, and F. This is not apparent in the figure.

19. On line 187 the authors reference a loop that constricts the central channel. Although a loop is represented in Figure 1c (denoted as the $\alpha 5$ - $\alpha 6$) it is not clear if this is the loop being referred to. It would be beneficial to the reader to clarify this.

20. Lines 192-197 are speculative and should be moved to the discussion.

21. On lines 199-203 the authors describe the electrostatic surface of Cag β in relation to TrwB. It would be helpful if the authors included an image showing the electrostatic character of TrwB for comparison.

22. On lines 225, the authors reference an experiment that they conducted using three different constructs of the AAD domain. It is not clear to the reader why these three constructs were selected.

23. On line 234 the authors describe the two molecules in the asymmetric unit as being 'essentially identical'. The authors should provide an RMSD for the two copies.

24. On line 248 the authors describe the total buried surface area in the Cag β -ADD-CagZ complex but do not describe how it was determined. A computational program should be cited.

25. Line 255 describes contributions from two residues (L196 and L198) to the Cag β -CagZ interface. The authors should specify which proteins these residues are from to avoid confusion.

26. There are problems in formatting for references 11, 25, 29, 42, and 44.

27. Reference 53, as cited on line 360, does not refer to the H. pylori type four secretion system.

28. In Figure 6, the authors present ATP hydrolysis data to enumerate the hydrolytic activity of Cag β . While error bars are represented on the graphs, it is not described in the methods or figure legend how the errors were calculated.

29. In Extended Data 4a and b the authors present several cropped images of SDS-PAGE gels of their purified mutants. The authors should include full uncropped images so that the reader can accurately assess the purity of these samples.

We thank the reviewers for their constructive critics and suggestions. We have carried out additional experiments to address some of the major concerns, including mass photometry experiments to confirm the conversion of Cag β between the monomeric and hexameric states, and mutational analyses to support our assignment of the ATPase active site residues. In response to the comments of Reviewer 3 regarding the poor quality of some of the ITC data, we have replaced these data with new ones. We have also reprocessed all the ITC data and changed the presentation of the errors in Table 1, which led to some small changes to the numbers but the conclusions remain the same. The following are point-by-point response to the reviewers' comments.

Reviewer #1 (Remarks to the Author):

In this manuscript, Wu et al. describe crystal structures of the *Helicobacter pylori* ATPase Cagbeta and a complex formed between a fragment of Cagbeta and CagZ, a protein that interacts with Cagbeta and regulates the translocation of the oncoprotein CagA through the type 4 secretion system into gastric epithelial cells where it dysregulates cell signaling in numerous ways that can lead to peptic ulcer diseases and gastric cancer. The authors show that full-length Cagbeta forms a hexameric ring structure and that the Cagbeta fragment bound to CagZ forms a 1:1 complex that is incompatible with hexameric formation. Furthermore, they performed activity assays to show that the ATPase activity of Cagbeta is inhibited in a dose-dependent manner by CagZ, but not a mutant of CagZ that does not bind to Cagbeta. The data is quite compelling and its presentation is generally fine. A few major points of concern follow:

Response: We thank the reviewer for these positive comments on the importance of this work and the quality of the manuscript.

1. Much of the authors' conclusions rest on the structure of the Cagbeta fragment-CagZ complex and the size exclusion chromatography analysis of full-length Cagbeta in the presence and absence of CagZ. In this crystal structure, there are two notable features: first, the Cagbeta is a fragment and, thus, not the natural full-length Cagbeta protein but an arbitrarily truncated portion; second, the C-terminal tail of CagZ in this structure adopts a conformation that is different from that of the apo CagZ structure and, moreover, makes numerous interactions with the Cagbeta fragment, which also undergoes conformational changes relative to its apo structure. Whether these conformational changes in both proteins and the resulting intermolecular interactions are caused by crystallization or not is unclear. Some solution biophysics could help here. There are numerous options, but performing a hydrogen-deuterium exchange-mass spectrometry experiment with the proteins in the crystal structure, as well as with full-length Cagbeta, would show both the interfaces gained by the interactions of these proteins and the interfaces lost in the proposed CagZ-mediated transition of full-length Cagbeta from hexameric to monomeric forms. These HDX-MS experiments, as well as additional analytical ultracentrifuge experiments, would also greatly corroborate the SEC data suggesting that CagZ transitions Cagbeta from hexamer to monomer (SEC is not the highest fidelity readout of oligomeric state).

Response: We agree with the above comments and thank the reviewer for the insightful suggestions. HDX-MS is a highly specialized technique, which we cannot do on our own. It would cause much delay even if we could manage to find a collaborator to do these experiments. A crystal structure of full length cag β in complex with CagZ would be great to have. In fact, we have tried co-expression and co-crystallization of Cag β ₁ (full-length of

the soluble region) with CagZ. Unfortunately, all attempts of crystallization of this complex failed. It is likely that, while the AAD is stabilized in the Cag β_1 /CagZ complex, the NBD and the NBD-AAD interdomain hinge become flexible due to lack of the stabilization effect of the hexameric assembly, rendering the complex refractory to crystallization.

Regarding the conformational changes of CagZ upon binding to Cag β , in the apo-CagZ structure published 2004¹, the authors predicted that the disordered C-terminal tail and the charged patch of the CagZ surface participate in protein/protein interactions¹. Our structure of the complex is consistent with these previous analyses. In addition, our extensive mutational data support the critical role of the CagZ tail in binding to Cag β . As discussed in the manuscript, there is no evidence suggesting that the conformational changes to the AAD of Cag β in the structure make major contributions to the binding to CagZ or regulation of Cag β in general. We therefore don't assign significance to these changes.

We agree with this reviewer that another method to confirm that the transition of Cag β from the hexamer to monomer upon binding CagZ is useful. In the revised manuscript, we address this point by using mass photometry, a new technique that can accurately measure the molecular weight of proteins^{2,3}. The results show that (revised Fig. 5d) Cag β_1 at 1.80 μ M is mostly hexameric, with ~74% of protein showing a measured molecular weight of ~408 kDa, very close to the theoretical molecular weight of the hexamer of ~402 kDa. The hexamer population decreased to 22% when the protein concentration was lowered to 0.06 μ M, meanwhile the monomeric species (~70 kDa, close to the theoretical molecular weight of the monomer of ~67 kDa) became more prominent. The sample containing both CagZ and Cag β_1 showed molecular weight of ~88 kDa, consistent with the theoretical molecular weight of the 1:1 complex of ~90 kDa. These new data provide strong support for our monomer to hexamer transition model.

2. The authors' structure of full-length Cagbeta shows that its Walker A motifs responsible for ATP binding and hydrolysis are formed within a single Cagbeta protomer (in "cis"), rather than through the participation of residues from neighboring protomers (in "trans") that is common to many Cagbeta homologs. This begs the question: why would CagZ-mediated trapping of a monomeric Cagbeta state inhibit ATP hydrolysis if a monomer still contains a whole Walker A motif? That is, what are the CagZ-driven changes, conformational or otherwise, in the Cagbeta Walker A motif that explain the related dysfunction of the ATPase I? Again, HDX-MS could indicate conformational and/dynamic changes in the Cagbeta Walker A motifs induced by CagZ.

Response: This is an important point, which we admit that we do not fully understand at present. As mentioned in the paper, most if not all the ATPases in this family function as hexamers, with the catalytic site residing at the inter-subunit interface. We reasoned that one important role of the hexameric interface is to help the Walker A and Walker B motifs adopting the catalysis-competent conformation. We agree that the arginine finger in cis in Cag β is unusual and warrants further investigation. We therefore mutated R241 (R241A and R241K), the putative arginine finger residue and tested the effects of the mutations on the ATPase activity. The results show that these mutations completely abolish the ATPase activity, confirming that R241 is critical for catalysis. Furthermore, we mutated key residues in Walker A (K244A) and Walker B (E551A and E551Q). These mutations also abolish the ATPase activity. These new data, included in revised Figure 6a, support our model on the catalytic mechanism.

Some minor comments follow:

1. There are a number of instances in which the order of figures and/or figure panels appear out of order with the text (e.g., figure 2 is described in text after the early panels of figure 1 but before the latter panels of figure 1; figure 7b is described before 7a).

Response: We thank the reviewer for pointing out these errors. We have made corrections by changing the order of the figures and the text.

2. Line 204: "biochemical assays" is vague.

Response: We have replaced this term with the specific techniques used (Size-exclusion chromatography and cross-linking assays).

3. Line 208: "superstable" - is this a technical term?

Response: We agree that this is not an accurate term. We now simply describe the hexamer as not very stable.

Reviewer #2 (Remarks to the Author):

This manuscript reports an exciting advance in the T4SS field concerning the structural definition of the Cagb coupling protein and effects of the CagZ adaptor structure and catalytic activity. The Cagb structure is only the second X-ray structure of the T4CP components of T4SSs, which allowed for detailed comparisons between Cagb and the current structural prototype for T4CPs, TrwB. The T4CPs are essential for translocation of DNA and protein substrates through most T4SSs, therefore, it is critical to know their architectures and effects of modulatory proteins such as CagZ. The apostructure of CagZ was previously reported, and here the authors show that CagZ undergoes a profound conformational change when bound to the all-alpha-domain (AAD) of Cagb. Previous studies have shown that T4CP AADs are important receptor domains for secretion substrates, and here the authors propose that CagZ coordinates binding of the secretion substrate CagA with the Cagb T4CP for translocation through the Cag T4SS. CagZ locks Cagb into the monoflate and also blocks its ATPase activity, suggesting that CagZ binding renders Cagb inactive. Their findings prompt a model in which the binding of CagA to the CagZ-Cagb complex results in dissociation of CagZ, which in turn allows Cagb hexamer formation and catalytic activity as a prerequisite for Cagb-mediated unfolding and translocation of CagA. Overall, the work is well-described and appears technically sound although this reviewer is not an X-ray crystallographer. It represents a major advance in the field insofar as there is no prior information describing in atomic detail the structural effects of an adaptor on the nucleotide binding domain of a T4CP. There are a number of grammatical flaws that can be easily rectified. As mentioned below, the authors should consider their model in the context of the recent CryoET structure of the Cag machine in the *H. pylori* envelope, which shows densities potentially corresponding to the Cagb hexamer at the base of the central channel. Other issues are minor.

Response: We thank this reviewer for these positive comments.

1. Pg. 5, L. 139 Walker. Fix other grammatical errors throughout.

Response: Thanks for spotting this error. We have thoroughly checked the manuscript to correct this and other errors.

2. Pg. 6. L. 150. This isn't evident from the presented structures, which isn't that important, but if the authors want to highlight this point, a different color scheme should be used for the C-terminal domain.

Response: We agree and therefore have deleted this sentence.

3. Pg. 6. L. 155. Not evident since there's no side-by-side comparison in this figure. There is a comparison in the suppl figure 1a,b, but this also does not reveal the size difference.

Response: Sorry for this oversight. We have added a new panel showing the side-by-side comparison of the two hexamers to illustrate the size difference (Revised Extended Data Figure 1).

4. Pg. 7. L. 226. Since no information can be gained for residues 260-460, it is possible that these residues also bind CagZ even though the AAD also does.

Response: Our structure of the AAD/CagZ complex shows that residues 297-476 constitute the AAD required for binding CagZ. Residues 260-460 are therefore indeed mostly the AAD. This question might be caused by our poor writing of this paragraph. We have made changes to it and hopefully the description has better clarity now.

5. Fig. 4 requires a fuller description in the legend about the data shown.

Response: Thanks for this suggestion. We have revised the figure legend accordingly.

6. Fig. 7. and Discussion. The model presented in Fig. 7 should be referred to early in the Discussion, even within the first paragraph. The model presents a flow of step-wise binding reactions that can be referred to throughout the Discussion.

Response: Thanks for these suggestions. We have added the reference to the model throughout the discussion.

7. The model is quite interesting, but needs to be reconciled with a couple of observations. First, a *deltacagZ* mutant lacks detectable CagA and, second, the CryoET structure shows densities consistent with a Cagb hexamer at the base of the inner ring. Relating to the *deltacagZ* phenotype, it isn't clear why *cagZ* binding to the Cagb monomer destabilizes CagA. Relating to the Cagb hexamer-IMC architecture detected by CryoET, this is assumed to represent the Cag machine in its quiescent state. If this is the case, it isn't obvious why Cagb assembles as the catalytically hexamer in the quiescent machine. Couldn't another model simply be that Cagb assembles as a hexamer but when docked onto the Cag machine it is catalytically inactive? CagZ could function as an adaptor by binding CagA and recruiting it to the AAD. Upon CagA-CagZ-Cagb binding, the CagZ – Cagb interaction induces conformational changes in the Cagb hexamer that do not necessarily induce monomerization but instead induces structural changes necessary for *dlf* CagA, an event that stimulates Cagb catalytic activity and, accordingly, CagA unfolding and translocation. This model is simpler and more consistent with results of the CryoET studies and CagZ-CagA biochemical studies. Regardless of whether the authors like this latter model, they need to do a better job reconciling their findings with the observations mentioned above.

Response: We appreciate this reviewer for these insightful comments. The actual processes of T4SS assembly and CagA transportation are likely much more complicated than reflected by the few structures presented by us and others. As pointed out by this reviewer, it is indeed intriguing that lack of CagZ causes loss of CagA. We speculate that CagZ is required for keeping the inactive pool of Cag β , which in turn is required for timely assembly of T4SS when CagA needs to be translocated.

The presence of the Cag β hexamer in the presumably quiescent state in the CryoET structure is another interesting but not fully understood question. It is worthy pointing out that in the more recent high-resolution cryo-EM structure of T4SS encoded by the R388 plasmid, the TrwB/VirD4 ATPase is missing, indicating that it is not stably associated (Reference 53 in the manuscript). It is possible that the assembly, docking and activation of the ATPase hexamer is a multi-step process. The cryo-ET and cryo-EM structures may represent different intermediate structures. The ATPase may remain inactive even when the hexamer is docked to the base of the inner ring. The formation of the hexamer is necessary for the activation of the ATPase, but the activity is regulated by additional mechanisms that ensure it is only fully active when CagA engages Cag β . This idea is consistent with the model suggested by this reviewer, where CagA may promote the

ATPase activity by inducing a conformational change to Cag β . In this case, it is however unlikely CagZ remains bound to the AAD of Cag β , because its binding is not compatible with the formation of the hexamer based on our structures. That said, we cannot formally rule out the possibility that CagZ may bind the hexamer Cag β in a different mode at present.

We have revised the discussion section to incorporate these points inspired by this reviewer.

Reviewer #3 (Remarks to the Author):

In the manuscript entitled “Mechanism of regulation of the *Helicobacter pylori* Cag β ATPase by CagZ” Wu et al. detail the structural analysis of the *H. pylori* protein known as Cag β , an ATPase associated with the *H. pylori* Cag type four secretion system (T4SS). The function of Cag β is to aid in the transference of the oncogenic protein CagA, into host cells. In this manuscript, the authors detail two structures of Cag β , one in a hexameric assembly and one in the monomeric form bound to the accessory protein CagZ. The analysis described here highlights a regulatory mechanism for CagZ that suggests it prevents premature oligomerization of Cag β . The authors provide evidencels mechanism using isothermal titration calorimetry, ATPase activity assays, and size exclusion chromatography. Although the observation is noteworthy and the presented mechanism interesting, it should be noted that several critical pieces of data are missing from the manuscript. Coupled with a severe lack of clarity, it is suggested that this manuscript is not suitable for publication. Noted below are several issues that have been raised for this manuscript.

Response: We thank this reviewer for the constructive critics. We have extensively revised the manuscript and included new data, which we hope can address these concerns.

Major Issues

1. The authors make several claims about the binding site of ATP in Cag β but do not present a crystal structure with ATP bound. In fact, most of the analysis concerning the ATP binding site is dependent on the placement of a SO₄²⁻ ion in the active site. Without a structure of Cag β bound to ATP, these comments are speculative. It is suggested that the authors work towards a co-crystal structure of Cag β in complex with ATP, or a non-hydrolysable analog.

Response: We agree that a crystal structure with ATP or a non-hydrolysable analog would be a valuable addition. We indeed tried repeatedly but failed to obtain high quality crystals for solving such a structure. As an alternative, we carried out mutational studies of key residues in the catalytic site according to our structural analyses. The results show that mutations of R241 (catalytic base), K244 (Walker A) and E551 (Walker B) abolish the ATPase activity of Cag β (revised Figure 6a). These new data lend strong support to our assignment of the catalytic site residues.

2. The evaluation of the ATP binding site is interesting as the authors highlight the lack of an arginine finger between adjacent protomers in the hexameric assembly and suggest that another residue (Arg241) takes its place in cis. The authors have clearly established an ATP hydrolysis assay, and as such, it leatly benefit their analysis to test active site mutants for ATPase activity.

Response: This is a great suggestion. As mentioned above, our new results show that both the R241A and R241K mutations abolish the ATPase activity (revised Figure 6a).

3. Crystallographic interactions are indicated in several figures throughout the manuscript, but no distances are given. At present, it is not clear to the reader how these interfaces have been determined. The authors should define this in the manuscript and label these interactions in their figures.

Response: We have added distance labels as suggested (see revised Figure 1h,i, Figure 3c.)

4. The authors have submitted several preliminary PDB validation reports for the Cag β structure that are all clearly marked “not for manuscript review”. The authors should provide a full, official validation report with their submission.

Response: We provide the official reports in this revised submission.

5. The authors present derived K_d values for a library of CagZ peptides binding to Cag β -AAD but do not present any data for these experiments. The data for these experiments must be included in the manuscript.

Response: We have included these data in the revised Figure 4d. We rearranged this figure and added more labels for clarity.

6. The authors present binding data for CagZ to Cag β (Extended Data Figure 5a), but the data appear to be very noisy. At present, it is hard to understand how the affinity for this interaction was determined and to what degree the readers can be confident in the reported K_d. The authors should provide additional experimental support for the determined K_d or remove this analysis from the manuscript.

Response: Thanks for this suggestion. We repeated these experiments, and now present the new data of much improved quality (revised Extended Data Figure 5a). The k_d between CagZ and Cag β , from these experiments is ~0.05 μ M, similar with that between CagZ and the isolated AAD (0.09 μ M). Prompted by these comments, we have also reprocessed all the ITC data and changed the presentation of the error range of k_d, which led to some small changes to the numbers but the conclusions from these results remain the same.

7. The authors present size exclusion chromatography data describing the oligomerization state of Cag β at high concentrations. As presented, it is unclear if the shift in retention volume is consistent with an ~6-fold shift in molecular weight. The authors should provide additional validation to support this claim.

Response: We agree with this concern. In the revised manuscript, we address this point by using mass photometry, a new technique that can accurately measure the molecular weight of proteins^{2,3}. Please see the response to point 1 of Reviewer 1 for details. The new data are presented in Fig. 5d, which confirm the conversion between the monomeric to hexameric state.

8. The authors provide crosslinking data to demonstrate the oligomerization of Cag β at high concentrations. As presented, the data are unclear, and it seems that most of the protein is aggregating when cross-linked. The authors should present a calibrated molecular size marker to indicate the molecular weight of the species that is formed in the presence of DSS.

Response: We have added the molecular size marker in the revised Extended data Figure 3. The crosslinking assay shows β_1 has the ability to form higher oligomer in the presence of crosslinking assay.

Minor Issues

1. Line 50: 'strains of *H. pylori* use their T4SS' should directly reference the Cag T4SS as *H. pylori* contains up to four T4SSs in total.

Response: Done as suggested.

2. Line 84 references the T4SS in *H. pylori*; again, the authors should specify that the T4SS referenced is the Cag T4SS.

Response: Done as suggested.

3. Line 107 is confusing as written. It is assumed that the authors mean that the determined structure consists of residues 166-748, though it implies that the transmembrane domain is formed by residues 166-748. The authors should also reference Figure 1a here.

Response: Great suggestions. We have revised the text to avoid the confusion and reference Figure 1a as suggested.

4. Extended data Figure 1a – the gray coloring of TrwB is difficult to see as presented. It would be best to change the color of TrwB to better illustrate the observed differences.

Response: thanks for this suggestion. We have changed the color of TrwB to magenta for better illustration.

5. It is suggested that the plot in Figure 1c be rotated.

Response: We prefer to keep this figure as is because this orientation is consistent with that of panel c.

6. In line 123 and Figure 1b; the figure does not clearly illustrate the authors comment concerning the fold of the NBD. The figure or the text should be reworked to better illustrate this point.

Response: Thanks for pointing out this problem. We labeled all the second structure elements in Figure 1b in the revised manuscript. This will help to illustrate the fold of NBD.

7. In Figure 2, residues forming the Walker A and/or B motifs should be noted in the figure to better orient the reader.

Response: Great suggestion. We have made changes to Figure 2 accordingly.

8. In line 127, the Walker A motif is referred to as 'Walk A'. This happens several times in the same paragraph. These need to be corrected.

Response: We appreciate this reviewer for the careful reading of our manuscript. We have thoroughly checked and corrected the mistakes in the revised manuscript.

9. In line 137, Walker A should be capitalized.

Response: Done.

10. In line 130, the authors describe density in the nucleotide binding site but show only the model. The referenced density should be shown in the figure.

Response: Thanks for your suggestion. We have added the density of sulfate in revised Figure 2b.

11. In lines 134-136 the authors argue that Arg241 of Cag β serves the same function as Arg375 of TrwB. The authors should show an overlay of the two active sites to illustrate this point.

Response: Thanks for this suggestion. We have added the overlay of the ATP binding site of Cag β and TrwB in revised Extended Data Figure 2b

12. In lines 142-143 the authors describe residues 631-658 as forming ‘the base layer of the 3-layered Cag β ’. This is confusing. Are the authors referring to the AAD, NBD, and B-barrel? It is also not clear what ‘base layer’ means in this context.

Response: Sorry for the vague description. We describe the hexamer of Cag β as having a three-layered structure: the top layer is composed of AAD, the middle layer is NBD, and bottom layer is formed by the β -barrel. We agree the term “base layer” might be misleading. We have changed it to “bottom layer”. We have labeled the three layers in revised Figure 1b and a side-by-side hexamer comparison of Cag β with TrwB in revised Extended Data Figure 1.

13. It would be useful to the readers if the authors showed a comparison between the β -strands of Cag β with those observed in TrwB.

Response: We have added the hexamer comparison between Cag β and TrwB with the β -strands labeled in revised Extended data Figure 1c, d.

14. In Figure 1d the authors show a view of all orientations of Cag β that are observed in the hexameric form. The authors should elaborate on how the AAD changes from one protomer to another. Does it rotate or translate and to what degree?

Response: Great suggestion. We have added an inset panel in revised Figure 1d to address this point. We chose residue 374 in each protomer of the AAD and residue 704 in each protomer of the NBD as the reference points to measure the distance between neighboring subunits. It is clear from the figure that the distances between neighboring AAD are quite different from one another. For example, the distance between AADs of subunits E and F (45.7Å) is much larger than that between subunits A and B (36.9Å). In contrast, the same analysis of the NBD shows that they are related to an approximate 6-fold symmetry. Therefore, the deviation from the 6-fold symmetry is largely due to shifts of AAD relative to NBD. The actual shifts of each AAD appear somewhat random and may be influenced by crystal contacts, and therefore we do not to describe them in detail.

15. On line 157 the authors reference the larger structure of Cag β compared to TrwB, but they do not illustrate this in the referenced figure (Figure 1e).

Response: We have added a side-by-side comparison of Cag β and TrwB, see revised Extended data Figure 1c, d.

16. On line 159 the authors reference the ATP binding site at the inter-subunit interface and reference Figure 1h. It is not clear where the proposed ATP binding site is in this figure.

Response: we now show the bound SO $_4^{2-}$ ion at the active site in Figure 1h, which helps pinpoint the active site.

17. In Figure 1g the authors intended to show the lack of interactions between adjacent AAD domains within the hexameric structure. This is not clearly shown. In fact, it looks like the authors even highlight a single interaction. It would help to demonstrate the distances observed between residues.

Response: The AAD domains make very few contacts in the hexameric structure. Figure 1g shows the AAD protomers from subunits A and B, which are closer each other than other AAD protomers. As seen the figure, the contacts are minimal even between the A and B subunits. The highlighted distance in the original Figure 1g is the polar contact between E350 from protomer A and T514 from protomer B. Distances between other residues from the two subunits are all larger than 4 Å, suggesting no direct interaction. To avoid confusion, we deleted the distance label in the revised Figure 1g.

18. In Figure 1d the authors reference the positions of the AADs in the hexamer and note that protomers A, B, and C have different orientations than D, E, and F. This is not apparent in the figure.

Response: Please see the response above. The new inset of Figure 1d shows the differences in distance between neighboring subunits.

19. On line 187 the authors reference a loop that constricts the central channel. Although a loop is represented in Figure 1c (denoted as the $\alpha 5$ - $\alpha 6$) it is not clear if this is the loop being referred to. It would be beneficial to the reader to clarify this.

Response: the constriction loop is between $\alpha 5$ and $\alpha 6$. We have labeled the secondary structure elements in Figure 1b and pointed out the $\alpha 5$ - $\alpha 6$ loop in the revised Figure 1 and manuscript.

20. Lines 192-197 are speculative and should be moved to the discussion.

Response: We have moved these sentences to the discussion.

21. On lines 199-203 the authors describe the electrostatic surface of Cag β in relation to TrwB. It would be helpful if the authors included an image showing the electrostatic character of TrwB for comparison.

Response: Great suggestion. We have added the side-by-side comparison in revised Extended Figure 1e, f.

22. On lines 225, the authors reference an experiment that they conducted using three different constructs of the AAD domain. It is not clear to the reader why these three constructs were selected.

Response: We did not know the exact domain boundaries of the AAD in Cag β . These experiments allowed us to map the boundaries, and choose the best construct for the crystallization experiments.

23. On line 234 the authors describe the two molecules in the asymmetric unit as being 'essentially identical'. The authors should provide an RMSD for the two copies.

Response: The RMSD for the two copies is 0.11 Å. This information is included in the revised manuscript.

24. On line 248 the authors describe the total buried surface area in the Cag β -ADD-CagZ complex but do not describe how it was determined. A computational program should be cited.

Response: The buried surface area was calculated using PDBe PISA server. This information is now included in the revised manuscript.

25. Line 255 describes contributions from two residues (L196 and L198) to the Cag β -CagZ interface. The authors should specify which proteins these residues are from to avoid confusion.

Response: L196 and L198 are from CagZ. We have included this information in the revised manuscript.

26. There are problems in formatting for references 11, 25, 29, 42, and 44.

Response: Fixed.

27. Reference 53, as cited on line 360, does not refer to the H. pylori type four secretion system.

Response: Thanks for pointing out this error. We have deleted this reference.

28. In Figure 6, the authors present ATP hydrolysis data to enumerate the hydrolytic activity of Cag β . While error bars are represented on the graphs, it is not described in the methods or figure legend how the errors were calculated.

Response: Thanks for pointing out this problem. The reported values are averages from three replicates, with the error bars indicating the standard deviations. We have added the description in the revised Figure 6 legend.

29. In Extended Data 4a and b the authors present several cropped images of SDS-PAGE

gels of their purified mutants. The authors should include full uncropped images so that the reader can accurately assess the purity of these samples.

Response: Uncropped gel images are now included (revised Extended Data 4a).

- 1 Cendron, L., Seydel, A., Angelini, A., Battistutta, R. & Zanotti, G. Crystal structure of CagZ, a protein from the *Helicobacter pylori* pathogenicity island that encodes for a type IV secretion system. *J Mol Biol* **340**, 881-889, doi:10.1016/j.jmb.2004.05.016 (2004).
- 2 Sonn-Segev, A. *et al.* Quantifying the heterogeneity of macromolecular machines by mass photometry. *Nat Commun* **11**, 1772, doi:10.1038/s41467-020-15642-w (2020).
- 3 Young, G. *et al.* Quantitative mass imaging of single biological macromolecules. *Science* **360**, 423-427, doi:10.1126/science.aar5839 (2018).

REVIEWERS' COMMENTS

Reviewer #1 (Remarks to the Author):

The authors have adequately addressed my critiques and the manuscript should be accepted for publication.

Reviewer #2 (Remarks to the Author):

In their revision, the authors have satisfactorily addressed my previous concerns. The other reviewers will weigh in, but my assessment is that the authors have worked hard to address those concerns, in some cases by providing additional data and in others by providing reasonable explanations for why certain requested data, e.g., presenting a crystal structure of the full-length versions of both proteins, cannot be generated at this time or in the reasonable near future.

Reviewer #3 (Remarks to the Author):

In the revised manuscript, Wu et al. present additional data in support of the conclusions made in their original submission. Notably, this includes the addition of mass photometry data which directly addresses a previous concern regarding the oligomerization state of CagBeta during this study. The authors have also provided additional ATP hydrolysis data for mutants contained within the proposed CagBeta Walker A/B motifs to support their assignments. Finally, the ITC data that were previously reported are now shown and new ITC data have been folded in to complete their analysis. With these revisions the manuscript is suitable for publication with the following minor revisions:

1. The authors present data that suggest that the CagBeta monomer contains the features necessary to interact with and hydrolyze ATP. In Figure 6a they also present data that demonstrate a concentration dependent ATP hydrolytic activity. The oligomeric state of CagBeta at these concentrations should be clearly stated in the text.
2. On lines 194-198 the authors mention the possibility that interactions between Arg241 and the adjacent protomer may be involved in regulating ATPase activity. It would help to define what these interactions are.
3. On lines 218 and 224 the authors reference experimental data summarized in table 2. These data should be included in the extended data and referenced.
4. The authors present ATP hydrolysis assays for CagBeta in the presence of at least three different constructs (CagZ, CagZ-24, and CagZ-20) and show that ATP hydrolysis activity is inhibited in the presence of only full length CagZ. It is assumed that CagZ-24 and CagZ-20 do not bind to CagBeta, but this has not been directly shown. The authors should revise the text to allow for the possibility that CagZ-20 and CagZ-24 still bind but do not inhibit.
5. Figure 7 and lines 376-380 included in the discussion reference a model that the authors are proposing involving CagBeta regulation. However, the inclusion of the cryo-ET data would suggest that the CagBeta does not need to interact with the CagA to oligomerize. The authors should revise this model to account for the incorporation of CagBeta into the T4SS in the absence of CagA.

Reviewer #1 (Remarks to the Author):

The authors have adequately addressed my critiques and the manuscript should be accepted for publication.

Reviewer #2 (Remarks to the Author):

In their revision, the authors have satisfactorily addressed my previous concerns. The other reviewers will weigh in, but my assessment is that the authors have worked hard to address those concerns, in some cases by providing additional data and in others by providing reasonable explanations for why certain requested data, e.g., presenting a crystal structure of the full-length versions of both proteins, cannot be generated at this time or in the reasonable near future.

Reviewer #3 (Remarks to the Author):

In the revised manuscript, Wu et al. present additional data in support of the conclusions made in their original submission. Notably, this includes the addition of mass photometry data which directly addresses a previous concern regarding the oligomerization state of CagBeta during this study. The authors have also provided additional ATP hydrolysis data for mutants contained within the proposed CagBeta Walker A/B motifs to support their assignments. Finally, the ITC data that were previously reported are now shown and new ITC data have been folded in to complete their analysis. With these revisions the manuscript is suitable for publication with the following minor revisions:

1. The authors present data that suggest that the CagBeta monomer contains the features necessary to interact with and hydrolyze ATP. In Figure 6a they also present data that demonstrate a concentration dependent ATP hydrolytic activity. The oligomeric state of CagBeta at these concentrations should be clearly stated in the text.

Response: While mass photometry can accurately measure the molecular weight and oligomerization states of proteins at optimal sample concentrations, it is difficult to obtain the distribution of different states across a wide range of concentrations. We therefore did not carry out mass photometry experiments of CagBeta at the concentration used in the ATPase assays in Figure 6. Data in Figure 6b and 6c show that the ATPase activity is inhibited by binding of CagZ, which, based on our structures, is incompatible with the hexamer of CagBeta but does not affect the ATPase active site. The results in Figure 6d show that the tail peptides of CagZ do not suppress the ATPase activity of CagBeta. These peptides can bind CagBeta (as described in lines 276-282 and the ITC results in Table 1), but do not impede the hexamer formation. We believe that these data together provide strong support for the model that the ATPase activity of CagBeta is dependent on the hexamer, while CagZ inhibits the activity by preventing its formation. Please also see the response to point 4 below.

2. On lines 194-198 the authors mention the possibility that interactions between Arg241 and the adjacent protomer may be involved in regulating ATPase activity. It would help to define what these interactions are.

Response: We have added two sentence to describe these interactions: “The sidechain of Arg241 makes contacts with several residues from the neighboring subunit, such as Tyr228, Glu570 and Glu676 (Fig. 1h). The detailed interactions made by Arg241 are different among the six subunits, due to the deviation from the 6-fold symmetry of the hexamer, which again may reflect the different conformations of the arginine finger needs to adopt in the catalytic cycle (Supplementary Fig. 2b).”

3. On lines 218 and 224 the authors reference experimental data summarized in table 2. These data should be included in the extended data and referenced.

Response: We have included these data in Supplementary Figure 4.

4. The authors present ATP hydrolysis assays for CagBeta in the presence of at least three different constructs (CagZ, CagZ-24, and CagZ-20) and show that ATP hydrolysis activity is inhibited in the presence of only full length CagZ. It is assumed that CagZ-24 and CagZ-20 do not bind to CagBeta, but this has not been directly shown. The authors should revise the text to allow for the possibility that CagZ-20 and CagZ-24 still bind but do not inhibit.

Response: Sorry for the confusion. In fact, these peptides do bind CagBeta, albeit with reduced affinity, as escribed in lines 276-282 and the ITC results in Table 1. As discussed in the answer to point 1 and lines 320-326 in the manuscript, the fact that these peptides can bind CagBeta but do not affect the ATPase activity (even when used at 100-fold excess) support our model that CagZ inhibits the CagBeta activity by preventing the formation of the hexamer. In contrast, the tail peptides bind the outer surface of CagBeta that is not involved in the hexamer formation as shown by the crystal structures, therefore has no impact on the formation of the CagBeta hexamer.

5. Figure 7 and lines 376-380 included in the discussion reference a model that the authors are proposing involving CagBeta regulation. However, the inclusion of the cryo-ET data would suggest that the CagBeta does not need to interact with the CagA to oligomerize. The authors should revise this model to account for the incorporation of CagBeta into the T4SS in the absence of CagA.

Response: This is a valid point. As it is difficult to convey this idea in the cartoon model without making it too complicated, we added a sentence at the end of the paragraph in the discussion to address this point “On the other hand, it remains possible that other factors may trigger the dissociation of CagZ and allow the

hexamer formation of Cag β in the absence of CagA.”